# Classification of Heavy-tailed Features in High Dimensions: a Superstatistical Approach

**Urte Adomaityte**
Department of Mathematics
King's College London
urte.adomaityte@kcl.ac.uk

**Gabriele Sicuro**
Department of Mathematics
King's College London
gabriele.sicuro@kcl.ac.uk

**Pierpaolo Vivo**
Department of Mathematics
King's College London
pierpaolo.vivo@kcl.ac.uk

## Abstract

We characterise the learning of a mixture of two clouds of data points with generic centroids via empirical risk minimisation in the high dimensional regime, under the assumptions of generic convex loss and convex regularisation. Each cloud of data points is obtained via a double-stochastic process, where the sample is drawn from a Gaussian distribution whose variance is itself a random parameter sampled from a scalar distribution $\varrho$. As a result, our analysis covers a large family of data distributions, including the case of power-law-tailed distributions with no covariance, and allows us to test recent "Gaussian universality" claims. We study the generalisation performance of the obtained estimator, we analyse the role of regularisation, and we analytically characterise the separability transition.

## 1 Introduction

Generalised linear models (GLMs) are still ubiquitous in the theoretical research on machine learning, despite their simplicity. Their nontrivial phenomenology is often amenable to complete analytical treatment and has been key to understanding the unexpected behavior of large and complex architectures. Random features models [61, 47, 24], for example, allowed to clarify many aspects of the well-known double-descent phenomenon in neural networks [54, 8, 9]. A line of research spanning more than three decades [68, 74] has considered a variety of GLMs to investigate a number of aspects of learning in high dimensions. Yet, a crucial assumption adopted in many such theoretical models is that the covariates are obtained from a *Gaussian* distribution, or from a mixture of Gaussian distributions [42, 49, 3, 37, 43]. Although such a Gaussian design has served as a convenient working hypothesis (in some cases experimentally and theoretically justified [67, 44]), it is however not obvious how much it limits the scope of the results. It is reasonable to expect structure, heavy tails, and large fluctuations in real (often non-Gaussian) data [1, 62] to play an important role in the learning process, and it would be therefore desirable to include structured and heavy-tailed-distributed covariates in our theoretical toolbox.

This paper presents, to the best of our knowledge for the first time, the exact asymptotics for classification tasks on covariates obtained from a mixture of heavy-tailed distributions. We focus on supervised binary classification assuming that the sample size $n$ and the dimensionality $d$ of the space where the covariates live are both sent to infinity, keeping their ratio $n/d = \alpha$ fixed. The paper fits therefore in the line of works on exact high-dimensional asymptotics for classification learning via a GLM [49, 43], but, crucially, we relax the usual Gaussian data hypothesis by including in our

37th Conference on Neural Information Processing Systems (NeurIPS 2023).

analysis power-law-tailed distributions with possibly no covariance. The mixture is obtained from two distributions, each centered around a centroid $\boldsymbol{\mu} \in \mathbb{R}^d$ and resulting from a double stochastic process. Namely, each sample point is obtained from a distribution $\mathcal{N}(\boldsymbol{\mu}, \boldsymbol{\Sigma})$ whose covariance $\boldsymbol{\Sigma} = \Delta \boldsymbol{I}_d \in \mathbb{R}^{d \times d}$ is itself a random variable so that $\Delta$ has density $\varrho$, supported on $\mathbb{R}_*^+$. Using, for example, an appropriately parametrised inverse-Gamma distribution for $\varrho$, such a *superstatistical* construction (as known in the physics literature [5, 7]) can provide, for example, a heavy-tailed, Cauchy-like data distribution with infinite covariance. The replica method [48], an analytical tool widely adopted in statistical physics, provides asymptotic formulas for a generic density $\varrho$, allowing us to study the effects of non-Gaussianity on the performance curves, and test Gaussian universality hypotheses [59] in such a classification task.

**Motivation and related works**   Learning a rule to classify data points clustered in clouds in high dimension is a classical problem in statistics [32]. Its ubiquity is exemplified by the recently observed neural collapse phenomenon in deep neural networks, in which the last layer classifier was found to operate on features clustered in clouds around the vertices of a simplex [38, 56]; more recently, Seddik et al. [67] showed that Gaussian mixtures suitably describe the deep learning representation of GAN data. In theoretical models, data points are typically assumed to be organised in *K Gaussian* clouds, each corresponding to a class. Each class $k$, $k \in \{1, \ldots, K\}$, is centered around a mean vector $\boldsymbol{\mu}_k$ and has covariance $\boldsymbol{\Sigma}_k$. The binary classification case, $K = 2$, is the simplest, and possibly most studied, setting. For this case, Mai and Liao [46] considered an ERM task with generic convex loss and ridge regularisation in the high-dimensional regime $n, d \rightarrow +\infty$ with $n/d \in (0, +\infty)$. In this setting, they gave a precise prediction of the classification error, showing the optimality of the square loss in the unregularised case. Their results have been extended by Mignacco et al. [49], who showed that the presence of a regularisation can actually drastically affect the performance, improving it. In the same setting, the Bayes optimal estimator has been studied in both the supervised and the semi-supervised setting [49, 41]; almost-Bayes-optimal solutions have been put in relation to wide flat landscape regions [3]. In the context of binary classification, the maximum number $n$ of samples that can be perfectly fitted by a linear model in dimension $d$ [49, 16, 37] has been the topic of investigation since the seminal work of Cover [12] and is related to the classical storage capacity problems on networks [20, 21, 39]. The corresponding separability transition value $\alpha = n/d$ is remarkably associated with the existence transition of the maximum likelihood estimator [70, 76]. The precise asymptotic characterisation of the test error in learning to classify $K \geq 2$ Gaussian clouds with generic means and covariances has been recently obtained by Loureiro et al. [43]. Within this line of research, rigorous results have been obtained by a variety of methods, such as Gordon's inequality technique [31, 71, 49] or mapping to approximate message passing schemes [4, 36, 10, 43, 27].

As previously mentioned, the working hypothesis of *Gaussian design* is widely adopted in high-dimensional classification problems. On top of being a convenient technical hypothesis, this assumption has been justified in terms of a "Gaussian universality" principle. In other words, in a number of circumstances, non-Gaussian features distributions are found to be effectively described by Gaussian ones with matching first and second moments as far as the asymptotic properties of the estimators are concerned [44]. Such a "universality property" has been rigorously proven for example in compressed sensing [50] or in the case of LASSO with non-Gaussian dictionaries [55]. A Gaussian equivalence principle introduced by Goldt et al. [30] has been proven to hold for a wide class of generalised linear estimation problems [47]. Extending work by Hu and Lu [34], Montanari and Saeed [51] recently proved such a principle in a GLM under the assumption of *pointwise normality* of the distribution of the features. This crucial assumption can be intended, roughly speaking, as the assumption of sub-Gaussian decay of a marginal of the feature distribution in any direction (see also [25, 13] for further developments).

The Gaussian universality principle, however, can break down by relaxing some of the aforementioned assumptions on the feature distribution. Montanari and Saeed [51] for example showed that pointwise normality is a *necessary* hypothesis for universality. Studying regression tasks on an elliptically distributed dataset, El Karoui [18] showed that claims of "universality" obtained in the Gaussian setting require serious scrutiny, as the statistics of the estimators might strongly depend on the non-Gaussianity of the covariates [18, 69, 72]. More recently, Pesce et al. [59] showed that a structured Gaussian mixture cannot be effectively described by a single Gaussian cloud. Building upon a series of contributions related to the asymptotic performance of models in the proportional high-dimensional limit [33, 47, 28, 24, 30, 44], we aim precisely to explore the validity of Gaussian universality within classification problems.

In this paper, we work in a "superstatistical" data setting [5, 7], meaning that we superpose the statistics of Gaussian data distribution with an assumed distribution of its variance. Such a construction is adopted in a number of disciplines and contexts to go beyond Gaussianity, albeit is known under different names. In statistical physics, it is known as "superstatistics" and is employed in the analysis of non-equilibrium and non-linear systems [6]. In Bayesian modeling, it is common to refer to hierarchical priors and models [22, 23], while in probability, statistics, and actuarial sciences such distributions are known as compound probability distributions [63], or doubly-stochastic models [60, 66]. Crucially, this construction allows us to consider a very large family of non-Gaussian distributions, which include, but are not limited to, any power-law decay and Cauchy-like with possible infinite-covariance parametrisations.

**Our contributions**    In this manuscript we provide the following results.

• We study a classification task on a non-Gaussian mixture model (see Eq. (1) below) via a generalised linear model (GLM) and we analytically derive, using the replica method [14, 19, 48], an asymptotic characterisation of the statistics of the empirical risk minimisation (ERM) estimator. Our results go therefore beyond the usual Gaussian assumption for the dataset, and include for example the case of covariates obtained from a mixture of distributions with infinite variance. The analysis is performed in the high-dimensional, proportional limit and for any convex loss and convex regularisation. By using this result, we provide asymptotic formulas for the generalisation, training errors and training loss.

• We analyse the performance of this ERM task on a specific family of dataset distributions by using different convex loss functions (quadratic and logistic) and ridge regularisation. We show in particular that, in the case of two balanced clusters with a specific non-Gaussian distribution, the optimal ridge regularisation strength $\lambda^\star$ is finite, at odds with the Gaussian case, for which $\lambda^\star \to \infty$ [49]. In this respect, by considering distributions with matching first and second moments, we analytically show that the performances of the analysed GLM do depend in general on higher moments, and therefore the "Gaussian universality principle" breaks down when heavy-tailed distributions are considered.

• We derive the separability threshold on a large family of non-Gaussian dataset distributions, possibly with unbounded covariance, generalising the known asymptotics for the separability of Gaussian clouds [49]. The result of Cover [12] is recovered in the case of infinite distribution width.

• Under some moment conditions, we derive the Bayes-optimal performance in the case of binary classifications with symmetric centroids, generalising the argument in Ref. [49].

• We finally extend recent results on Gaussian universality in the Gaussian mixture model with random labels [25, 59] to the case of non-Gaussian distributions. We show that the universal formula for the square training loss at zero regularisation found in Ref. [25] holds in our more general setting as well.

## 2   Main result

**Dataset construction**    We consider the task of classifying two data clusters in the $d$-dimensional space $\mathbb{R}^d$. The dataset $\mathcal{D} \coloneqq \{(\boldsymbol{x}^\nu, y^\nu)\}_{\nu \in [n]}$ is obtained by extracting $n$ independent datapoints $\boldsymbol{x}^\nu$, each associated with a label $y^\nu \in \{-1, 1\}$ (a more general setting, involving a multiclass classification task, is discussed in Appendix A). The data points are correlated with the labels via a law $P(\boldsymbol{x}, y)$ which we assume to have the form

$$P(\boldsymbol{x}, y) = \delta_{y,1}\rho P(\boldsymbol{x}|\boldsymbol{\mu}_+) + \delta_{y,-1}(1-\rho)P(\boldsymbol{x}|\boldsymbol{\mu}_-), \qquad \rho \in (0,1), \qquad \boldsymbol{\mu}_\pm \in \mathbb{R}^d. \tag{1a}$$

Here $P(\boldsymbol{x}|\boldsymbol{\mu})$ is a distribution with mean $\boldsymbol{\mu}$, and the mean vectors $\boldsymbol{\mu}_\pm \in \mathbb{R}^d$ are distributed according to some density, such that $\mathbb{E}\left[\|\boldsymbol{\mu}\|^2\right] = \Theta(1)$, and correspond to the center of the two clusters. The scalar quantity $\rho$ weighs the relative contribution of the two clusters: in the following, we will denote $\rho_+ = \rho = 1 - \rho_-$. Each cluster distribution $P(\boldsymbol{x}|\boldsymbol{\mu})$ around a vector $\boldsymbol{\mu}$ is assumed to have the form

$$P(\boldsymbol{x}|\boldsymbol{\mu}) \coloneqq \mathbb{E}_\Delta\left[\mathcal{N}\left(\boldsymbol{x}\,|\boldsymbol{\mu}, \Delta \boldsymbol{I}_d\right)\right], \tag{1b}$$

where $\mathcal{N}(\boldsymbol{x}|\boldsymbol{\mu}, \boldsymbol{\Sigma})$ is a Gaussian distribution with mean $\boldsymbol{\mu} \in \mathbb{R}^d$ and covariance $\boldsymbol{\Sigma} \in \mathbb{R}^{d \times d}$, and $\Delta$ is randomly distributed with some density $\varrho$ with support on $\mathbb{R}_*^+ \coloneqq (0, +\infty)$. The family of "elliptic-like" distributions in Eq. (1b) has been extensively studied, for instance, by the physics community, in the context of *superstatistics* [5, 7]. Mixtures of normals in the form of Eq. 1b are a central tool in Bayesian statistics [65] due to their ability to approximate any distribution given a sufficient number

of components [53, 2, 29]. Although the family in Eq. (1b) is not the most general of such mixtures, it is sufficient to include a large family of power-law-tailed densities and to allow us to go beyond the usual Gaussian approximation. El Karoui [18] considered the statistical properties, in the same asymptotic proportional regime as here, of ridge-regularised regression estimators on datasets with distribution as in Eq. 1b, under the assumption (here relaxed) that $\mathbb{E}[\Delta^2] < +\infty$. This *elliptic* family includes a large class of distributions with properties markedly different from the ones of Gaussians. For example, the inverse-Gamma with density $\varrho(\Delta) = (2\pi\Delta^3)^{-1/2}\,e^{-\frac{1}{2\Delta}}$ leads to a $d$-dimensional Cauchy-like distribution $P(x|\mu) \propto (1 - \|\mu - x\|^2)^{-\frac{d+1}{2}}$, having as marginals Cauchy distributions and $\mathbb{E}[\|x - \mu\|^2] = +\infty$. We will perform our classification task by searching for a set of parameters $(w^\star, b^\star)$, called *weights* and *bias*, respectively, that will allow us to construct an estimator via a certain classifier $\varphi\colon \mathbb{R} \to \{-1, 1\}$

$$x \mapsto \varphi\left(\frac{x^\top w^\star}{\sqrt{d}} + b^\star\right). \tag{2}$$

By means of the above law, we will predict the label for a new, unseen data point $x$ sampled from the same law $P(x, y)$. Our analysis will be performed in the high-dimensional limit where both the sample size $n$ and dimensionality $d$ are sent to infinity, with $n/d \equiv \alpha$ kept constant.

**Learning task**   In the most general setting, the parameters are estimated by minimising an empirical risk function in the form

$$(w^\star, b^\star) \equiv \arg\min_{\substack{w \in \mathbb{R}^d \\ b \in \mathbb{R}}} \mathcal{R}(w, b) \quad \text{where} \quad \mathcal{R}(w, b) \equiv \sum_{\nu=1}^{n} \ell\left(y^\nu, \frac{w^\top x^\nu}{\sqrt{d}} + b\right) + \lambda r(w). \tag{3}$$

Here $\ell$ is a strictly convex loss function with respect to its second argument, and $r$ is a strictly convex regularisation function with the parameter $\lambda \geq 0$ tuning its strength.

**State evolution equations.**   Let us now present our main result, namely the exact asymptotic characterisation of the distribution of the estimator $w^\star \in \mathbb{R}^d$ and of $1/\sqrt{d}w^{\star\top}X \in \mathbb{R}^n$, where $X \in \mathbb{R}^{d \times n}$ is the concatenation of the $n$ dataset column vectors $x^\nu \in \mathbb{R}^d$, $\nu \in [n]$. Such an asymptotic characterisation is performed via a set of order parameters satisfying a system of self-consistent "state-evolution" equations, which we will solve numerically. The asymptotic expressions for generalisation and training errors, in particular, are written in terms of their solutions, the order parameters. In the following, we use the shorthand $\ell_\pm(u) \equiv \ell(\pm 1, u)$, and given a function $\Phi\colon \{-1, 1\} \to \mathbb{R}$, we write $\Phi_\pm \equiv \Phi(\pm 1)$ and $\mathbb{E}_\pm[\Phi_\pm] := \rho_+\Phi_+ + \rho_-\Phi_-$.

**Result 2.1** *Let $\zeta \sim \mathcal{N}(0, 1)$ and $\Delta \sim \varrho$, both independent from other quantities. Let also be $\xi \sim \mathcal{N}(\mathbf{0}, I_d)$. In the setting described above, given $\phi_1\colon \mathbb{R}^d \to \mathbb{R}$ and $\phi_2\colon \mathbb{R}^n \to \mathbb{R}$, the estimator $w^\star$ and the vector $z^\star := \frac{1}{\sqrt{d}}w^\star X \in \mathbb{R}^n$ verify:*

$$\phi_1(w^\star) \xrightarrow[n,d\to+\infty]{P} \mathbb{E}_\xi\left[\phi_1(g)\right], \qquad \phi_2(z^\star) \xrightarrow[n,d\to+\infty]{P} \mathbb{E}_\zeta\left[\phi_2(h)\right], \tag{4}$$

*where we have introduced the proximal $g \in \mathbb{R}^d$, defined as*

$$g := \arg\min_w \left(\hat{v}\frac{\|w\|^2}{2} - \sqrt{d}\sum_{k=\pm}\hat{m}_k w^\top \mu_k - \sqrt{\hat{q}}\xi^\top w + \lambda r(w)\right) \tag{5}$$

*and where we also have introduced the proximal for the loss $h \in \mathbb{R}^n$, obtained by concatenating $\rho_+ n$ quantities $h_+$ with $\rho_- n$ quantities $h_-$, with $h_\pm$ given by*

$$h_\pm := \arg\min_u \left[\frac{(u-\omega_\pm)^2}{2\Delta v} + \ell_\pm(u)\right], \qquad \text{where } \omega_\pm := m_\pm + b + \sqrt{q\Delta}\zeta. \tag{6}$$

*The collection of parameters $(q, m_\pm, v, \hat{q}, \hat{m}_\pm, \hat{v}, b)$ appearing in the equations above is given by the fixed-point solution of the following self-consistent equations:*

$$\begin{cases} m_\pm = \frac{1}{\sqrt{d}}\mathbb{E}_\xi\left[g^\top\mu_\pm\right], \\ q = \frac{1}{d}\mathbb{E}_\xi[\|g\|^2], \\ v = \frac{1}{d}\hat{q}^{-1/2}\mathbb{E}_\xi[g^\top\xi], \end{cases} \begin{cases} \hat{q} = \alpha\mathbb{E}_{\pm,\zeta,\Delta}\left[\Delta f_\pm^2\right], \\ \hat{v} = -\alpha q^{-1/2}\mathbb{E}_{\pm,\Delta,\zeta}\left[\sqrt{\Delta}f_\pm\zeta\right], \\ \hat{m}_\pm = \alpha\rho_\pm\mathbb{E}_{\Delta,\zeta}[f_\pm], \end{cases} \quad \begin{aligned} f_\pm &:= \frac{h_\pm - \omega_\pm}{v\Delta}, \\ b &= \mathbb{E}_{\pm,\Delta,\zeta}[h_\pm - m_\pm]. \end{aligned} \tag{7}$$

The derivation of the result above is given, in a multiclass setting, in Appendix A using the replica method, which can be put on rigorous ground by a mapping of the risk minimisation problem into an approximate message-passing iteration scheme [17, 43, 26]. As in the purely Gaussian case, the state evolution equations naturally split the dependence on the loss function $\ell$, which is relevant in the computation of $h_\pm$, and on the regularisation function $r$, entering in the computation of $g$. In the most general setting, we are required to solve two convex minimisation problems, consisting of the computation of the proximals $h_\pm$ and $g$ [57]; the convexity of the problem guarantees the uniqueness of the fixed point solution. Result 2.1 implies the following.

**Result 2.2** *Under the assumptions of Result 2.1, the training loss $\epsilon_\ell$, the training error $\epsilon_t$, and the test (or generalisation) error $\epsilon_g$ are given, in the proportional asymptotic limit, by*

$$\frac{1}{n}\sum_{\nu=1}^{n}\ell\left(y^\nu,\frac{w^{\star\top}x^\nu}{\sqrt{d}}+b^\star\right)\to\mathbb{E}_{\pm,\zeta,\Delta}[\ell_\pm(h_\pm)]=:\epsilon_\ell,$$

$$\frac{1}{n}\sum_{\nu=1}^{n}\mathbb{I}\left(\varphi\left(\frac{w^{\star\top}x^\nu}{\sqrt{d}}+b^\star\right)\neq y^\nu\right)\to\mathbb{E}_{\pm,\zeta,\Delta}[\mathbb{I}(\varphi(h_\pm)\neq\pm1)]=:\epsilon_t, \tag{8}$$

$$\mathbb{E}_{(y,x)}\left[\mathbb{I}\left(\varphi\left(\frac{w^{\star\top}x}{\sqrt{d}}+b^\star\right)\neq y\right)\right]=\mathbb{E}_{\pm,\Delta,\zeta}[\mathbb{I}(\varphi(\omega_\pm)\neq\pm1)]=:\epsilon_g.$$

**Result 2.3** *Under the hypothesis that $\mathbb{E}[\Delta]<+\infty$ and $\mathbb{E}[\Delta^{-2}]<+\infty$, the Bayes optimal test error for binary classification is*

$$\epsilon_g^{\mathrm{BO}}=\rho_+\mathbb{E}\left[\hat{\Phi}(\kappa_+)\right]+\rho_-\mathbb{E}\left[\hat{\Phi}(\kappa_-)\right],\qquad \kappa_\pm:=\frac{1}{\sqrt{\Delta_0^\star\left(1+\frac{\mathbb{E}[\Delta]\mathbb{E}[\Delta^{-2}]}{\alpha\mathbb{E}[\Delta^{-1}]^2}\right)}}\left(1\pm\frac{\Delta_0}{2}\left(1+\frac{1}{\alpha\mathbb{E}[\Delta^{-1}]}\right)\ln\frac{\rho_+}{\rho_-}\right) \tag{9}$$

*where the expectation is over the two identically distributed random variables $\Delta_0,\Delta_0^\star\sim\varrho$, and $\hat{\Phi}(x)$ is the complementary error function.*

**Quadratic loss with ridge regularisation.** Explicit choices for $\ell$ or $r$ can highly simplify the fixed-point equations above. As an example, let us consider the case of quadratic loss $\ell(y,x)=\frac{1}{2}(y-x)^2$ with ridge regularisation $r(w)=\frac{1}{2}\|w\|^2$. In this setting, an explicit formula for the proximal $h_\pm$ in Eq. (6) and for the proximal $g$ in Eq. (5) can be easily found. As a result, we can obtain explicit expressions for the corresponding set of state evolution equations which become particularly suitable for a fast numerical solution and explicit the dependence on higher moments of $\Delta$:

$$\begin{cases}m_\pm=\frac{\sum_{k=\pm}\hat{m}_k\mu_k^\top\mu_\pm}{\lambda+\hat{v}},\\ q=\frac{\|\sum_{k=\pm}\hat{m}_k\mu_k\|^2+\hat{q}}{(\lambda+\hat{v})^2},\\ v=\frac{1}{\lambda+\hat{v}},\end{cases}\qquad\begin{cases}\hat{q}=\alpha\mathbb{E}_\pm[(\pm1-m_\pm-b)^2]\mathbb{E}_\Delta\left[\frac{\Delta}{(1+v\Delta)^2}\right]+\alpha q\mathbb{E}_\Delta\left[\frac{\Delta^2}{(1+v\Delta)^2}\right],\\ \hat{v}=\alpha\mathbb{E}_\Delta\left[\frac{\Delta}{1+v\Delta}\right],\\ \hat{m}_\pm=\alpha\rho_\pm(\pm1-m_\pm-b)\mathbb{E}_\Delta\left[\frac{1}{1+v\Delta}\right].\end{cases} \tag{10}$$

## 3 Application to synthetic datasets

In this section, we compare our theoretical predictions with the results of numerical experiments for a large family of data distributions. The results have been obtained using ridge $\ell_2$-regularisation, and both quadratic and logistic losses, with various data cluster balances $\rho$. We will also assume, without loss of generality, that $\mu_\pm=\pm\frac{1}{\sqrt{d}}\mu$, where $\mu\sim\mathcal{N}(0,I_d)$. The synthetic data sets will be produced using, an inverse-Gamma-distributed variance $\Delta$, with density parametrised as

$$\varrho(\Delta)\equiv\varrho_{a,c}(\Delta)=\frac{c^a}{\Gamma(a)\Delta^{a+1}}\,e^{-\frac{c}{\Delta}}, \tag{11a}$$

depending on the shape parameter $a>0$ and on the scale parameter $c>0$. As a result, each class will be distributed as

$$P(x|\mu)=\frac{(2c)^a\Gamma(a+d/2)}{\Gamma(a)\pi^{d/2}(2c+\|x-\mu\|^2)^{a+d/2}}, \tag{11b}$$

which produces a cloud centered in $\mu$. The reason for our choice is that the distribution $\varrho_{a,c}$ allows us to easily explore different regimes by changing two parameters only. Indeed, the distribution has, for $a>1$, covariance matrix $\Sigma=\mathbb{E}[(x-\mu)\otimes(x-\mu)^\top]=\sigma^2I_d$, where $\sigma^2=\frac{c}{a-1}$. By fixing $\sigma^2$ and taking the limit $a\to+\infty$, $P(x|\mu)\to\mathcal{N}(x|\mu,\sigma^2I_d)$, i.e., the Gaussian case discussed by Mignacco et al. [49]. For $a\in(0,1]$, instead, $\sigma^2=+\infty$. Note that the choice of inverse-Gamma distribution is not that unusual. It has been adopted, for example, to describe non-Gaussian data in quantitative finance [15, 40] or econometrics models [52]. Finally, we will construct our label estimator as in Eq. (2) with $\varphi(x)=\mathrm{sign}(x)$.

## 3.1 Finite-covariance case

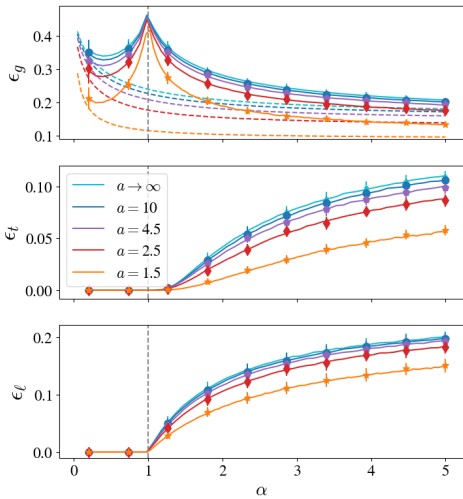

Figure 1: Test error $\epsilon_g$ (solid line, *top*), training error $\epsilon_t$ (*center*) and training loss $\epsilon_\ell$ (*bottom*) as predicted by Eq. (8) in the balanced $\rho = 1/2$ case. The dataset distribution is parametrised as in Eq. (12). The classification task is solved using a quadratic loss with ridge regularisation with $\lambda = 10^{-5}$. In the top figure, the dashed line corresponds to the Bayes optimal bound. Dots correspond to the average outcome of 50 numerical experiments in dimension $d = 10^3$. In our parametrisation, the population covariance is $\Sigma = I_d$ for all values of $a$ and moreover, for $a \to +\infty$, the case of Gaussian clouds with the same centroids and covariance is recovered. For further details on the numerical solutions, see Appendix B.

Let us start by considering a data distribution as in Eq. (11) with shape parameter $a = c + 1 > 1$, which is therefore given by

$$P(\boldsymbol{x}|\boldsymbol{\mu}) = \frac{2^a (a-1)^a \Gamma(a+d/2)}{\Gamma(a)\pi^{d/2}\left(2a-2+\|\boldsymbol{x}-\boldsymbol{\mu}\|^2\right)^{a+d/2}} \quad (12)$$

and decays as $\|\boldsymbol{x}\|^{-2a-1}$ in the radial direction for $\|\boldsymbol{x} - \boldsymbol{\mu}\| \gg 0$. As a consequence, the distribution has $\lim_d 1/d\,\mathbb{E}[\|\boldsymbol{x} - \boldsymbol{\mu}\|^k] = +\infty$ for $k \geq 2a$. With this choice, *all elements of the family in Eq.* (12) *have the same covariance* $\Sigma = \mathbb{E}[(\boldsymbol{x} - \boldsymbol{\mu}) \otimes (\boldsymbol{x} - \boldsymbol{\mu})^\top] = I_d$ *as $a$ is varied*, including the Gaussian limit $P(\boldsymbol{x}|\boldsymbol{\mu}) \to \mathcal{N}(\boldsymbol{\mu}, I_d)$ obtained for $a \to +\infty$.

In Fig. 1 we present the results of our numerical experiments using the square loss and small regularisation. An excellent agreement between the theoretical predictions and the results of numerical experiments is found for a range of values of $a$ and sample complexity $\alpha$, both for balanced, i.e., equally sized, and unbalanced clusters of data (the plot for this case can be found in Appendix A.3). The test error $\epsilon_g$ presents the classical interpolation peak at $\alpha = 1$, smoothened by the presence of a non-zero regularisation strength $\lambda$, and the typical double-descent behavior [54]. As $a$ is increased, the results of Mignacco et al. [49] for Gaussian clouds with $P(\boldsymbol{x}|\boldsymbol{\mu}) = \mathcal{N}(\boldsymbol{x}|\boldsymbol{\mu}, I_d)$ are approached. The $a \to +\infty$ curves correspond, therefore, to the Gaussian mixture model we would have constructed by simply fitting the first and second moments of each class in the finite-$a$ case. The plot shows that, at given population covariance $\Sigma = I_d$, the classification of power-law distributed clouds is associated with different, and in particular smaller, errors than the corresponding task on Gaussian clouds with the same covariance: in this sense, no Gaussian universality principle holds. This (perhaps) counter-intuitive effect stems from the fact that, although for all classes $\Sigma = I_d$, in our example the mean absolute deviation is smaller for finite $a$

$$\lim_{d \to +\infty} \frac{\mathbb{E}[\|\boldsymbol{x}-\boldsymbol{\mu}\|]}{\sqrt{\mathbb{E}[\|\boldsymbol{x}-\boldsymbol{\mu}\|^2]}} = \frac{\Gamma(a-1/2)\sqrt{a-1}}{\Gamma(a)} \xrightarrow{a \to +\infty} 1^-. \quad (13)$$

The learning process benefits from the fact that as $a$ decreases while keeping the variance fixed, points move on average closer to their means $\boldsymbol{\mu}_\pm$ to compensate heavier tails. Finally, we observed that test errors are systematically smaller for unbalanced clusters than for balanced clusters (see Appendix A.3).

The same numerical experiment was repeated using the logistic loss for training. Once again, in Fig. 2, we focus on $\rho = 1/2$ and show that the theoretical predictions of the generalisation and training errors agree with the results of numerical experiments for a range of sample complexity values $\alpha$ and various values of $a$. Just as for the square loss, the test error $\epsilon_g$ of the Gaussian model ($a \to +\infty$) is larger than the one observed for power-law distributed clouds at finite values of $a$. In the $\lambda \to 0$ limit, the typical interpolation cusp in the generalisation error is observed: the cusp occurs at different values of $\alpha$ as $a$ is changed; interpolation occurs at smaller values of $\alpha$ for larger $a$ (i.e., for "more Gaussian" distributions). A comparison with the test error $\epsilon_t$ confirms that this cusp occurs at the value of $\alpha$ where the training error becomes non-zero and the two training data clouds become non-separable [70, 16]. This sharp transition in $\alpha$ also corresponds to the existence transition of maximum-likelihood estimator in high-dimensional logistic regression, analysed for Gaussian data in Refs. [70, 49] and, in our setting, in Section 4 below. For larger regularisation strength, the cusp smoothens, and the training error becomes non-zero at smaller values of $\alpha$.

## 3.2 Infinite-covariance case

The family of distributions specified by Eq. (11) allows us to also consider the case of infinite covariance, i.e., $\sigma^2 = +\infty$. To test our formulas in this setting, we considered the case in which each cloud is obtained by "contaminating" a standard Gaussian with an infinite-covariance distribution as in Eq. (11b) with $c = 1$ and $a < 1$ [35]. In other words, we use the density $\varrho(\Delta) = r\varrho_{a,1}(\Delta) + (1-r)\delta(\Delta-1)$, with $r \in [0,1]$ interpolation parameter, for $\Delta$ in Eq. (1). Each class has therefore infinite covariance for $0 < r \leq 1$ and the Gaussian case is recovered for $r = 0$. Fig. 3 collects our results for two balanced clouds, analysed using square loss and logistic loss. Good agreement between the theoretical predictions and the results of numerical experiments is found for a range of values of sample complexity and of the ratio $r$. In this case, the finite-variance case $r = 0$ corresponds, not surprisingly, to the lowest test error with both square loss and logistic loss, which grows with $r$.

## 3.3 The role of regularisation in the classification of non-Gaussian clouds

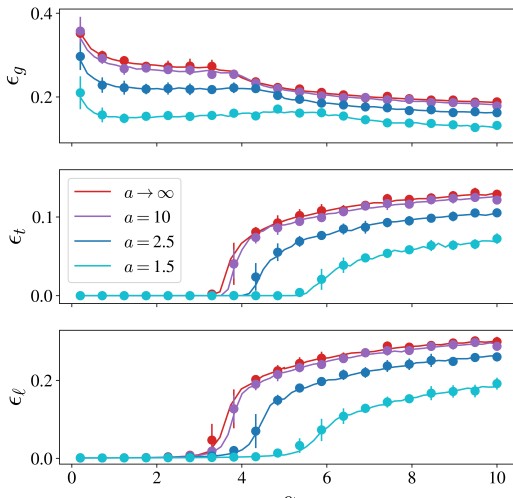

Figure 2: Test error $\epsilon_g$ (*top*), training error $\epsilon_t$ (*center*) and training loss $\epsilon_\ell$ (*bottom*) via logistic loss training on balanced clusters parametrised as in Eq. (12) ($\Sigma = I_d$). Ridge regularisation with $\lambda = 10^{-4}$ is adopted. Dots correspond to the average over 20 numerical experiments with $d = 10^3$. The Gaussian limit is recovered for $a \to +\infty$. Further details on the numerical solutions can be found in Appendix B.

One of the main results in the work of Mignacco et al. [49] is related to the effect of regularisation on the performances in a classification task on a Gaussian mixture. They observed that, for all values of the sample complexity $\alpha$, the optimal ridge classification performance on a pair of balanced clouds is obtained for an infinite regularisation strength $\lambda$, using both hinge loss and square loss. We tested the robustness of this result beyond the purely Gaussian setting. In Fig. 4 (left), we plot the test error obtained using square loss and different regularisation strengths $\lambda$ on the dataset distribution as in Eq. (12) with $a = 2$. We observe that, both in the balanced case and in the unbalanced case, the optimal regularisation strength $\lambda^\star$ is *finite* and, moreover, $\alpha$-dependent. On the other hand, if we fix $\alpha$ and let $a$ grow towards $+\infty$, the optimal $\lambda^\star$ grows as well to recover the result of Mignacco et al. [49] of diverging optimal regularisation for *balanced* Gaussian clouds, as shown in Fig. 4 (center)

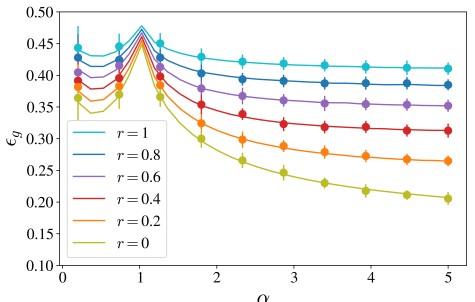
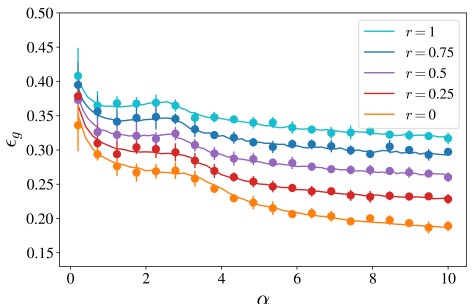

Figure 3: Test error $\epsilon_g$ in the classification of two balanced clouds, via quadratic loss (*left*) and logistic loss (*right*). In both cases, ridge regularisation is adopted ($\lambda = 10^{-4}$ for the square loss case, $\lambda = 10^{-3}$ for the logistic loss case). Each cloud is a superposition of a power-law distribution with infinite variance and a Gaussian with covariance $\Sigma = I_d$. The parameter $r$ allows us to contaminate the purely Gaussian case ($r = 0$) with an infinite-variance contribution ($0 < r \leq 1$) as in Eq. (11) with $c = 1$ and $a = 1/2$ (*left*) or $a = 3/4$ (*right*). Dots correspond to the average test error of 20 numerical experiments in dimension $d = 10^3$. Note that, at a given sample complexity, Gaussian clouds are associated with the lowest test error for both losses.

for $\alpha = 2$. This result is represented also in Fig. 4 (right), where it is shown that, in the *unbalanced* case, the optimal regularisation strength $\lambda^\star$ instead saturates to a finite value for $a \to +\infty$, i.e., for Gaussian clouds [49].

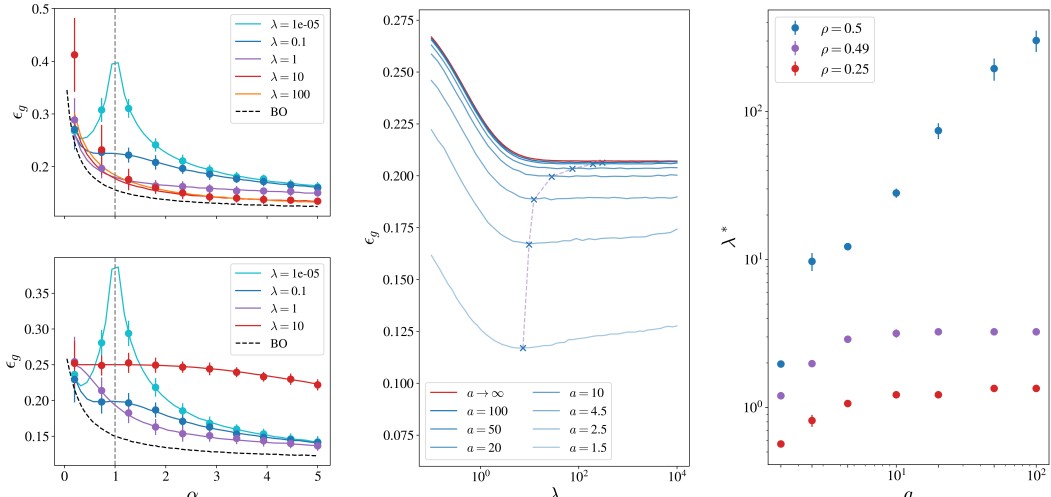

Figure 4: (*Left*) Test error for ridge regularised quadratic loss for various regularisation strengths. The data points of each cloud in the training set are distributed as in Eq. (12), with shape parameter $a = 2$, for balanced clusters (*top*) and unbalanced clusters ($\rho = 1/4$, *bottom*). Points are the results of 50 numerical experiments, and the dashed lines are Bayes-optimal bounds. (*Center*) Test error for different regularisation strengths $\lambda$ for two *balanced* clusters with quadratic loss at sample complexity $\alpha = 2$ using the data distribution (12). The optimal regularisation strength value $\lambda^\star$ obtained from averaging 5 runs for each $a$ is marked with a cross. (*Right*) Optimal regularisation strength $\lambda^\star$ at $\alpha = 2$ for different values of $a \in [1.5, 10^2]$ for both balanced and unbalanced clusters, obtained from averaging 5 runs. Note that, for $\rho = 1/2$, $\lambda^\star \to +\infty$ as $a \to +\infty$.

## 4 The separability threshold

By studying the training error $\epsilon_t$ with logistic loss at zero regularisation we can obtain information on the boundary between the regime in which the training data are perfectly separable and the regime in which they are not. In other words, we can extract the value of the so-called separability transition complexity $\alpha^\star$ [70, 49, 43]. Once again, a complete characterisation of this transition point is available in the Gaussian mixture case, where $\alpha^\star$ can be explicitly given as a function of $\sigma^2$ [49].

It is not trivial to extend this result to the general case of non-Gaussian mixtures. It is, however, possible to derive $\alpha^\star$ within the large family of "superstatistical" models we are considering here under the assumption that Result 2.1 holds. In this case, in Appendix A.3 we prove the following.

**Result 4.1** *In the considered double-stochastic model, data are linearly separable for $\alpha < \alpha^\star$, where*

$$\alpha^\star := \max_{\theta \in (0,1], \gamma} \frac{1 - \theta^2}{S(\theta, \gamma)}, \qquad S(\theta, \gamma) := \int_0^\infty f^2 \mathbb{E}_\Delta \left[ \rho_+ \mathcal{N}\left( f + \frac{\theta + \gamma}{\sqrt{\Delta}} \Big| 0, 1 \right) + \rho_- \mathcal{N}\left( f + \frac{\theta - \gamma}{\sqrt{\Delta}} \Big| 0, 1 \right) \right] df. \quad (14)$$

Fig. 5 shows the values of $\alpha^\star$ for different choices of the shape parameters $a$ and $c$ in the case of two balanced clouds of datapoints distributed as in Eq. (11b) as predicted by Eq. (14). In Fig. 5 (left), we fixed $\sigma^2 = \frac{c}{a-1} < +\infty$, and plotted the separability threshold $\alpha^\star$ for a range of values of the shape parameter $a > 1$. As $a$ grows, the known threshold value for the Gaussian mixture case is recovered [49]. The double limit $a \to \infty$ and $\sigma^2 \to \infty$, on the other hand, provides the Cover [12] transition value $\alpha^\star = 2$, as expected. In Fig. 5 (right), instead, we analyse the case of infinite-variance clusters, by fixing $a \in (0, 1)$ and by varying the scale parameter $c$ in Eq. (11b), which controls the spread (width) of the distribution. The Cover transition is therefore correctly recovered as $c$ diverges for all values of $a$.

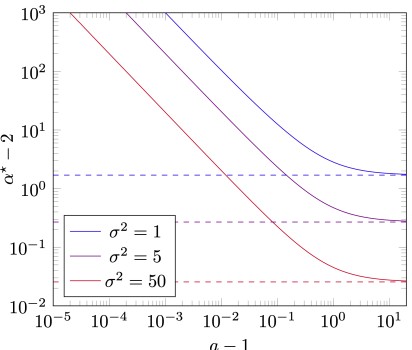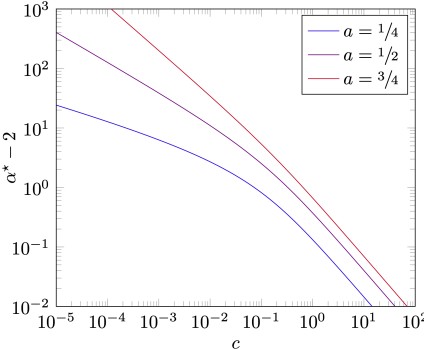

Figure 5: Separability threshold $\alpha^\star$ obtained by solving the equations in Eq. (7) with logistic loss, ridge regularisation strength $\lambda = 10^{-5}$ and $\rho = 1/2$. The data points of each cloud are distributed around their mean $\boldsymbol{\mu}$ with a two-parameter distribution as in Eq. (11b). (*Left*). Finite covariance $\boldsymbol{\Sigma} = \sigma^2 \boldsymbol{I}_d$ case, $\sigma^2 = \frac{c}{a-1}$, as a function of $a$. Dashed lines are the threshold values of the Gaussian case derived by Mignacco et al. [49]. At large $a$ and large variance $\sigma^2$, the Cover's transition $\alpha = 2$ for balanced clusters is recovered. (*Right*) Infinite-variance data clouds obtained by fixing $0 < a < 1$ in Eq. (11b).

## 5 Random labels in the teacher-student scenario and Gaussian universality

Our analysis can be extended to the case in which a point $\boldsymbol{x}$ is labeled not according to its class but by a "teacher", represented by a probabilistic law $P_0(y|\tau)$ and a vector $\boldsymbol{\theta}_0 \in \mathbb{R}^d$ so that each dataset element is independently generated with the joint distribution

$$P(\boldsymbol{x}, y) = P_0\left(y \middle| \frac{\boldsymbol{\theta}_0^\intercal \boldsymbol{x}}{\sqrt{d}}\right) \sum_{k=\pm} \rho_k \mathbb{E}[\mathcal{N}\left(\boldsymbol{x} \middle| \boldsymbol{\mu}_k, \Delta \boldsymbol{I}_d\right)]. \tag{15}$$

Let us assume now that $\boldsymbol{\theta}_0$ is such that $\lim_d 1/\sqrt{d}\,\boldsymbol{\theta}_0^\intercal \boldsymbol{\mu}_\pm = 0$, i.e., the teacher is "uncorrelated" with the structure of the mixture. In Appendix A.5 we show that, if $\ell(y, \eta) = \ell(-y, -\eta)$ and $P_0(y|\tau) = P_0(-y|-\tau)$, the task is equivalent to a classification problem on a *single* cloud centered in the origin, i.e., to the case $\boldsymbol{\mu}_\pm = \boldsymbol{0}$, so that $P(\boldsymbol{x}, y) = P_0\left(y \middle| 1/\sqrt{d}\,\boldsymbol{\theta}_0^\intercal \boldsymbol{x}\right) \mathbb{E}[\mathcal{N}(\boldsymbol{x}|\boldsymbol{0}, \Delta \boldsymbol{I}_d)]$. This mean-invariance, recently proven in Ref. [25] in the pure Gaussian setting, holds therefore in our non-Gaussian setting as well.

A special, relevant case of "uncorrelated teacher" is the one of *random labels*, where $P_0(y|\tau) = 1/2(\delta_{y,+1} + \delta_{y,-1})$. This apparently vacuous model is relevant in the study of the storage capacity problem [21, 39, 11] and of the separability threshold [12], and it has also been an effective tool to better understand deep learning [45, 75]. In this case, the following universal result holds with respect to our setup.

**Result 5.1** *In the random label setting studied via ridge regression case with zero regularisation, the asymptotic training loss is independent of the distribution of $\Delta$, and given by the universal formula*

$$\frac{1}{2n} \sum_{\nu=1}^\infty \left(y^\nu - \frac{\boldsymbol{x}^\intercal \boldsymbol{w}^\star}{\sqrt{d}}\right)^2 \xrightarrow[\substack{n/d = \alpha}]{n \to +\infty} \frac{\alpha-1}{2\alpha}\theta(\alpha-1) =: \epsilon_\ell. \tag{16}$$

The proof of the previous result is given in Appendix A.5. Note that this result does not require $\sigma^2 := \mathbb{E}[\Delta]$ to be finite, and it includes, as a special case, the Gaussian mixture model with random labels considered in Ref. [25, 59]. Therein, Eq. (16) has been obtained under the Gaussian design hypothesis. It turns out therefore that Eq. (16) holds for a much larger ensemble, which includes heavy-tailed distributions with possibly infinite covariance. On the other hand, universality *does not* hold, e.g., for the training error.

In Fig. 6 we exemplify these results by using ridge-regularised square loss. The numerical experiments correspond to a classification task on a dataset of *two* clouds centered in $\boldsymbol{\mu}_\pm$, $\boldsymbol{\mu}_+ = -\boldsymbol{\mu}_- \sim \mathcal{N}(\boldsymbol{0}, 1/d\boldsymbol{I}_d)$, obtained using the distribution

$$P(\boldsymbol{x}, y) = \frac{\delta_{y,+1} + \delta_{y,-1}}{4} \sum_{k=\pm} P(\boldsymbol{x}|\boldsymbol{\mu}_k), \tag{17}$$

where $P(x|\mu_\pm)$ is the distribution in Eq. (11b) depending on the shape parameters $a$ and $c$. These results are compared with the theoretical prediction obtained for a *single* cloud, with the same parameters $a$ and $c$, but centered in the origin, so that $P(x, y) = \frac{\delta_{y,+1}+\delta_{y,-1}}{2}P(x|\mu = 0)$: the agreement confirms the "mean-invariance" property stated above. As anticipated, as we are considering random labels, the training loss $\epsilon_\ell$ follows Eq. (16) in all cases, whereas the test and training errors are not universal. Note that, in place of the test error (which is trivially $\epsilon_g = 1/2$) we plot the mean square error on the pre-activations $\hat{\epsilon}_g := \mathbb{E}_{(y,x)}[(y - 1/\sqrt{a}x^\top w^\star)^2]$. Also, observe that this quantity is divergent in the case of infinite covariance, hence is absent from the plot for $a \in (0, 1]$.

## 6 Conclusions and perspectives

The framework introduced in this work has allowed us to perform an exact asymptotic analysis of the learning of a mixture of two data point clouds with possibly power-law tails. In our dataset, each sample point $x$ is labeled by $y \in \{-1, 1\}$ and obtained as $x = y\mu + \sqrt{\Delta}z$, where $\mu \in \mathbb{R}^d$, $z \sim \mathcal{N}(0, I_d)$ and $\Delta \sim \varrho$. We have derived exact expressions for the test and training errors and training loss achieved by a GLM in the limit of large size $n$ of the dataset and large dimensionality $d$ under the assumption that $\alpha = n/d$ is kept fixed. We have shown that the performance crucially depends on higher-order moments of the data distribution, so that a Gaussian universality assumption is not feasible. Moreover, we have shown the effects of non-Gaussianity on the optimal regularisation strength and derived an exact formula for the critical $\alpha^\star$ at which the two clouds become linearly inseparable. Finally, we have considered the so-called random labels setting (in which sample label assignment is randomised) and shown that, in this case, the training square loss takes a universal asymptotic form independent of the distribution of $\Delta$: this result further supports the crucial role of labels' randomness in some recent universality results [25].

In conclusion, we have provided an exactly solvable toy model exhibiting power-law tails and possibly unbounded covariance, highlighting the fact that a dataset deviation from Gaussianity in high dimensions cannot, in general, be neglected. The application of the introduced theoretical framework to real datasets is an interesting direction to explore in the future: the main difficulty, in this case, is the choice of the best distribution $\varrho$ given the observed dataset, a problem that has a long tradition in the context of Bayesian estimation [2, 23][1].

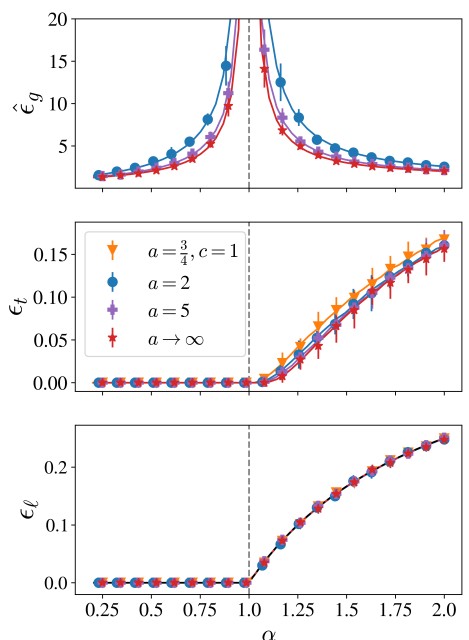

Figure 6: Mean square error $\hat{\epsilon}_g$ (*top*), training error $\epsilon_t$ (*center*) and training loss $\epsilon_\ell$ (*bottom*) obtained using square loss on a dataset distributed as in Eq (17) with random labels (dots) compared with the theoretical predictions for the single cluster case (continuous lines). The distributions are parametrised using Eq. (12) for $a > 1$ and Eq. (11b) for $0 < a \le 1$. A ridge regularisation with $\lambda = 10^{-4}$ is adopted. Note that $\hat{\epsilon}_g$ is absent in the top plot for $a = 1/2$ and $c = 1$ being a divergent quantity in this case.

## Acknowledgements

The authors would like to thank Nikolas Nüsken for stimulating discussions. Also, we would like to thank Bruno Loureiro for his feedback and for his careful reading of the manuscript.

[1]In a more simplistic approach, it can be observed that in the case in which the square loss is adopted, equations (10) depend on $\mathbb{E}[(1 + v\Delta)^{-1}]$ and $\mathbb{E}[(1 + v\Delta)^{-2}]$ only. Such quantities can be numerically estimated from the dataset, for example, by empirically evaluating the identity $\mathbb{E}[(1 + v\Delta)^{-1}] = \sqrt{v}\partial_v \int_0^v (v - u)^{-1/2}\mathbb{E}[\exp(-ux^2/2)]du$ when it is assumed that $x \sim \mathbb{E}[\mathcal{N}(0, \Delta)]$.

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
