# A Derivation of the fixed-point equations

In this Appendix, we will derive the fixed-point equations for the order parameters presented in the main text, following and generalising the analysis in Ref. [43]. The problem will be presented in a slightly more general setting then the one considered in the main text, namely considering a **multiclass classification** task on $K$ classes. The dataset $\mathcal{D} := \{(x^\nu, y^\nu)\}_{\nu \in [n]}$ we are going to consider consists of $n$ independent datapoints $x^\nu \in \mathbb{R}^d$ each associated with a label $y^\nu \in \mathcal{Y} \subseteq \mathbb{R}$, where $\mathcal{Y} = \{y_k\}_{k \in [K]}$ is a finite set of $K$ elements (for example, for $K = 2$ it is standard to choose $\mathcal{Y} = \{-1, 1\}$). The elements of the dataset are independently generated by using a law $P(x, y)$ which we assume can be put in the form $P(x, y) \equiv \int_0^\infty P(x, y|\Delta)\varrho(\Delta)d\Delta$ for some given distribution on the positive real axis $\varrho$, and such that

$$P(x, k|\Delta) = \rho_k \mathcal{N}\left(x \,\big|\, \mu_k, \Delta I_d\right), \qquad \rho_k \in (0, 1) \; \forall k \in \mathcal{Y}, \quad \sum_k \rho_k = 1. \tag{18}$$

The vectors $\mu_k \in \mathbb{R}^d$ play the role of centroids of each cluster $y = k \in \mathcal{Y}$. We will perform our classification task searching for a set of parameters $(w^\star, b^\star)$, called respectively *weights* and *bias*, that will allow us to construct an estimator via a certain classifier

$$x \mapsto \varphi\left(\frac{w^{\star\top}x}{\sqrt{d}} + b^\star\right) \in \mathcal{Y} \tag{19}$$

to estimate the label of a new, unobserved datapoint $x$. Here $\varphi \colon \mathbb{R} \to \mathcal{Y}$. To fix the ideas, in all our numerical experiments we used $K = 2$ and $\varphi(x) = \text{sign}(x)$. The choice of the parameters is performed by minimising an empirical risk function constructed via a loss function $\ell \colon \mathcal{Y} \times \mathbb{R} \to \mathbb{R}$ and a proper regularisation $r \colon \mathbb{R}^d \to \mathbb{R}$, in the form

$$\mathcal{R}(w, b) \equiv \sum_{\nu=1}^n \ell\left(y^\nu, \frac{w^\top x^\nu}{\sqrt{d}} + b\right) + \lambda r(w), \tag{20}$$

i.e., they are given by

$$(w^\star, b^\star) \equiv \arg\min_{\substack{w \in \mathbb{R}^d \\ b \in \mathbb{R}}} \mathcal{R}(w, b). \tag{21}$$

We will assume that the loss function $\ell$ is convex with respect to its second argument. The parameter $\lambda \geq 0$ tunes the strength of the regularisation $r$, which also is assumed to be convex. The starting point of our approach is the reformulation of the task as an optimisation problem by introducing a Gibbs measure over the parameters $(w, b)$ depending on a non-negative parameter $\beta$,

$$\mu_\beta(w, b) \propto e^{-\beta\mathcal{R}(w,b)} = \underbrace{e^{-\beta r(w)}}_{P_w(w)} \prod_{\nu=1}^n \underbrace{\exp\left[-\beta\ell\left(y^\nu, \frac{w^\top x^\nu}{\sqrt{d}} + b\right)\right]}_{P_y(y|w,b)}, \tag{22}$$

so that, in the $\beta \to +\infty$ limit, $\mu_\beta(w, b)$ concentrates on the values $(w^\star, b^\star)$ that minimize the empirical risk $\mathcal{R}(w, b)$ and are therefore the goal of the learning process. The functions $P_y$ and $P_w$ can be interpreted as (unnormalised) likelihood and prior distribution respectively. Our analysis will go through the computation of the average free energy density associated with this Gibbs measure in a specific proportional limit, i.e.,

$$f_\beta := -\lim_{\substack{n,d\to+\infty \\ n/d=\alpha}} \mathbb{E}_\mathcal{D}\left[\frac{\ln \mathcal{Z}_\beta}{d\beta}\right] = \lim_{\substack{n,d\to+\infty \\ n/d=\alpha}} \lim_{s\to 0} \frac{\ln \mathbb{E}_\mathcal{D}[\mathcal{Z}_\beta^s]}{sd\beta}, \qquad \mathcal{Z}_\beta := \int e^{-\beta\mathcal{R}(w,b)} \, dw. \tag{23}$$

where $\mathbb{E}_\mathcal{D}[\bullet]$ is the average over the training dataset. To perform the computation of such quantity, we use the so-called replica method.

## A.1 Replica approach

We proceed in our calculation by assuming no prior on $b$, which will play a role of an extra (low-dimensional) parameter whose optimal value will be derived extremising with respect to it the final result for the free energy. We need to evaluate

$$\mathbb{E}_\mathcal{D}[\mathcal{Z}_\beta^s] = \prod_{a=1}^s \int dw^a P_w(w^a) \left(\sum_k \rho_k \mathbb{E}_{x|k}\left[\prod_{a=1}^s P_k\left(\frac{w^a x}{\sqrt{d}} + b\right)\right]\right)^n. \tag{24}$$

Here and in the following, for the sake of brevity, $P_k(\eta) \equiv P_y(y_k|\eta)$ and similarly $\ell_k(\eta) \equiv \ell(y_k|\eta)$. Let us take the inner average introducing a new set of variables $\eta^a$,

$$\mathbb{E}_{\boldsymbol{x}|k}\left[\prod_{a=1}^{s} P_k\left(\frac{\boldsymbol{w}^{a\top}\boldsymbol{x}}{\sqrt{d}} + b\right)\right] = \prod_{a=1}^{s}\int \mathrm{d}\eta^a P_k(\eta^a)\mathbb{E}_{\boldsymbol{x}|k}\left[\prod_{a=1}^{s}\delta\left(\eta^a - \frac{\boldsymbol{w}^{a\top}\boldsymbol{x}}{\sqrt{d}} + b\right)\right]$$

$$= \mathbb{E}_{\Delta|k}\left[\prod_{a=1}^{s}\int \mathrm{d}\eta^a P_k(\eta^a)\mathcal{N}\left(\eta\left|\frac{\boldsymbol{w}^{a\top}\boldsymbol{\mu}_k}{\sqrt{d}} - b; \frac{\Delta \boldsymbol{w}^a \boldsymbol{w}^{b\top}}{d}\right.\right)\right], \quad (25)$$

where we used the superstatistical form of the distribution of a datapoint $\boldsymbol{x}$. Using the shorthand $\mathbb{E}_{k,\Delta}[\Phi_k(\Delta)] \equiv \sum_k \rho_k \int_0^\infty \varrho(\Delta)\Phi_k(\Delta)\mathrm{d}\Delta$, we can write then

$$\mathbb{E}_{\mathcal{D}}[\mathcal{Z}_\beta^s] = \prod_{a=1}^{s}\int \mathrm{d}\boldsymbol{w}^a P_w(\boldsymbol{w}^a)\left(\mathbb{E}_{k,\Delta}\left[\prod_{a=1}^{s}\int \mathrm{d}\eta^a P_k(\eta^a)\mathcal{N}\left(\boldsymbol{\eta}; \frac{\boldsymbol{w}^{a\top}\boldsymbol{\mu}_k}{d} + b; \frac{\Delta \boldsymbol{w}^a \boldsymbol{w}^{b\top}}{d}\right)\right]\right)^n$$

$$= \left(\prod_{a\leq b}\iint \mathcal{D}\boldsymbol{Q}^{ab}\mathcal{D}\hat{\boldsymbol{Q}}^{ab}\right)\left(\prod_a \iint \mathcal{D}\boldsymbol{m}^a\mathcal{D}\hat{\boldsymbol{m}}^a\right)\mathrm{e}^{-d\beta\Phi^{(s)}}. \quad (26)$$

In the equation above we introduced the *order parameters*

$$Q_\Delta^{ab} = \Delta\frac{\boldsymbol{w}^{a\top}\boldsymbol{w}^b}{d} \in \mathbb{R}, \quad a, b = 1, \ldots, s, \quad (27)$$

$$m_k^a = \frac{\boldsymbol{w}^{a\top}\boldsymbol{\mu}_k}{\sqrt{d}} \in \mathbb{R}, \quad a = 1, \ldots, s, \quad k \in \mathcal{Y}, \quad (28)$$

whilst $\iint \mathcal{D}\boldsymbol{Q}^{ab}\mathcal{D}\hat{\boldsymbol{Q}}^{ab}$ express the integration over $Q_\Delta^{ab}$ and $\hat{Q}_\Delta^{ab}$, to be intended as functions of $\Delta$. Similarly, $\mathcal{D}\boldsymbol{m}^a\mathcal{D}\hat{\boldsymbol{m}}^a \propto \prod_k \mathrm{d}m_k^a\mathrm{d}\hat{m}_k^a$. We have also introduced the replicated free-energy

$$\beta\Phi^{(s)}(Q, M, \hat{Q}, \hat{m}, b) = \sum_k \sum_a \hat{m}_k^{a\top}m_k^a + \sum_{a\leq b}\mathbb{E}_\Delta[\hat{Q}_\Delta^{ab}Q_\Delta^{ab}]$$

$$- \frac{1}{d}\ln\prod_{a=1}^{s}\int P_w(\boldsymbol{w}^a)\mathrm{d}\boldsymbol{w}^a\left(\prod_{a\leq b}\mathrm{e}^{\mathbb{E}_\Delta[\Delta\hat{Q}_\Delta^{ab}]\boldsymbol{w}^{a\top}\boldsymbol{w}^b}\prod_a \mathrm{e}^{\sqrt{d}\sum_k \hat{m}_k^a\boldsymbol{w}^{a\top}\boldsymbol{\mu}_k}\right)$$

$$- \alpha\ln\mathbb{E}_{k,\Delta}\left[\prod_{a=1}^{s}\int \mathrm{d}\eta^a P_k(\eta^a)\mathcal{N}\left(\boldsymbol{\eta}|m_k^a + b, Q_\Delta^{ab}\right)\right]. \quad (29)$$

At this point, the free energy $f_\beta$ should be computed functionally extremizing with respect to all the order parameters by virtue of the Laplace approximation (in addition to $b$),

$$f_\beta = \lim_{s\to 0}\operatorname*{Extr}_{\substack{m,\hat{m}\\Q,\hat{Q},b}}\frac{\Phi^{(s)}(Q, m, \hat{Q}, \hat{m}, b)}{s}. \quad (30)$$

However, the convexity of the problem allows us to make an important simplification.

**Replica symmetric ansatz** Before taking the $s \to 0$ limit we make the replica symmetric assumptions

$$Q_\Delta^{aa} = \begin{cases} R_\Delta, & a = b \\ Q_\Delta & a \neq b \end{cases} \qquad \hat{Q}_\Delta^{aa} = \begin{cases} -\frac{1}{2}\hat{R}_\Delta, & a = b \\ \hat{Q}_\Delta & a \neq b \end{cases} \quad (31)$$

$$m_k^a = m_k \qquad\qquad \hat{m}_k^a = \hat{m}_k \quad \forall a$$

By means of the replica symmetric hypothesis, we can write

$$Q_\Delta^{ab} \mapsto \boldsymbol{Q}_\Delta \equiv (R_\Delta - Q_\Delta)\boldsymbol{I}_{s,s} + Q_\Delta\boldsymbol{1}_s. \quad (32)$$

The inverse matrix is therefore

$$\boldsymbol{Q}_\Delta^{-1} = \frac{1}{R_\Delta - Q_\Delta}\boldsymbol{1}_s - \frac{Q_\Delta}{(R_\Delta - Q_\Delta + sQ_\Delta)(R_\Delta - Q_\Delta)}\boldsymbol{I}_{s,s}, \quad (33)$$

whereas

$$\det \mathbf{Q}_\Delta = (R_\Delta - Q_\Delta)^{s-1}(R_\Delta - Q_\Delta + sQ_\Delta) = 1 + s\ln(R_\Delta - Q_\Delta) + s\frac{Q_\Delta}{R_\Delta - Q_\Delta} + o(s). \quad (34)$$

If we denote $V_\Delta := R_\Delta - Q_\Delta$

$$\ln \mathbb{E}_{k,\Delta}\left[\prod_{a=1}^s \int d\eta^a P_k(\eta^a)\mathcal{N}\left(\eta\middle|m_k^a + b, Q_\Delta^{ab}\right)\right] = s\mathbb{E}_{k,\Delta,\zeta}\left[\ln Z_k\left(m_k + b + \sqrt{Q_\Delta}\zeta, V_\Delta\right)\right] + o(s), \quad (35)$$

with $\zeta \sim \mathcal{N}(0,1)$ normally distributed random variable. In the expression above, we have also introduced the function

$$Z_k(m,V) := \int \frac{d\eta P_k(\eta)}{\sqrt{2\pi V}} e^{-\frac{(\eta-m)^2}{2V}}. \quad (36)$$

On the other hand, denoting by $\hat{V}_\Delta = \hat{R}_\Delta + \hat{Q}_\Delta$,

$$\frac{1}{d}\ln\prod_{a=1}^s\left(\int d\boldsymbol{w}^a P_w(\boldsymbol{w}^a)e^{-\mathbb{E}_\Delta[\Delta\hat{V}_\Delta]\frac{\|\boldsymbol{w}^a\|^2}{2}+\sqrt{d}\sum_k \hat{m}_k \boldsymbol{w}^{a\top}\boldsymbol{\mu}_k}\prod_b e^{\frac{1}{2}\mathbb{E}_\Delta[\Delta\hat{Q}_\Delta]\boldsymbol{w}^{a\top}\boldsymbol{w}^b}\right) =$$

$$= \frac{s}{d}\mathbb{E}_\xi\ln\left[\int d\boldsymbol{w} P_w(\boldsymbol{w})\exp\left(-\mathbb{E}_\Delta[\Delta\hat{V}_\Delta]\frac{\|\boldsymbol{w}\|^2}{2}+\sqrt{d}\sum_k \hat{m}_k\boldsymbol{w}^\top\boldsymbol{\mu}_k+\sqrt{\mathbb{E}_\Delta[\Delta\hat{Q}_\Delta]}\boldsymbol{\xi}^\top\boldsymbol{w}\right)\right] + o(s). \quad (37)$$

In the expression above we have introduced $\boldsymbol{\xi} \sim \mathcal{N}(\boldsymbol{0}, \boldsymbol{I}_d)$ and denote the average over it by $\mathbb{E}_{\boldsymbol{\xi}}[\bullet]$. Therefore, the (replicated) *replica symmetric* free-energy is given by

$$\lim_{s\to 0}\frac{\beta}{s}\Phi_{\text{RS}}^{(s)} = \sum_k \hat{m}_k m_k + \frac{\mathbb{E}_\Delta\left[\hat{V}_\Delta Q_\Delta - \hat{Q}_\Delta V_\Delta - \hat{V}_\Delta V_\Delta\right]}{2} - \alpha\beta\Psi_{\text{out}}(m,Q,V) - \beta\Psi_w(\hat{m},\hat{Q},\hat{V}) \quad (38)$$

where we have defined two contributions

$$\Psi_{\text{out}}(m,Q,V) := \beta^{-1}\mathbb{E}_{k,\Delta,\xi}\left[\ln Z_k(\omega_k, V_\Delta)\right]$$

$$\Psi_w(\hat{m},\hat{Q},\hat{V}) := \frac{1}{\beta d}\mathbb{E}_\xi\ln\left[\int P_w(\boldsymbol{w})d\boldsymbol{w}\exp\left(-\frac{\mathbb{E}_\Delta[\Delta\hat{V}_\Delta]\|\boldsymbol{w}\|^2}{2}+\sqrt{d}\sum_k \hat{m}_k\boldsymbol{w}^\top\boldsymbol{\mu}_k+\sqrt{\mathbb{E}_\Delta[\Delta\hat{Q}_\Delta]}\boldsymbol{\xi}^\top\boldsymbol{w}\right)\right] \quad (39)$$

and introduced, for future convenience,

$$\omega_k := m_k + b + \sqrt{Q_\Delta}\zeta. \quad (40)$$

Note that we have separated the contribution coming from the chosen loss (the so-called *channel* part $\Psi_{\text{out}}$) from the contribution depending on the regularisation (the *prior* part $\Psi_w$). To write down the saddle-point equations in the $\beta \to +\infty$ limit, let us first rescale our order parameters as $\hat{m}_k \mapsto \beta\hat{m}_k$, $\hat{Q}_\Delta \mapsto \beta^2\hat{Q}_\Delta$, $\hat{V} \mapsto \beta\hat{V}$ and $V_\Delta \mapsto \beta^{-1}V_\Delta$. For $\beta \to +\infty$ the channel part is

$$\Psi_{\text{out}}(m,Q,V) = -\mathbb{E}_{k,\Delta,\zeta}\left[\frac{(h_k-\omega_k)^2}{2V_\Delta} + \ell_k(h_k)\right]. \quad (41)$$

where we have written $\Psi_{\text{out}}$ in terms of a proximal

$$\arg\min_u\left[\frac{(u-\omega_k)^2}{2V_\Delta} + \ell_k(u)\right]. \quad (42)$$

A similar expression can be obtained for $\Psi_w$. Introducing the proximal

$$\boldsymbol{g} = \arg\min_{\boldsymbol{w}}\left(\frac{\mathbb{E}_\Delta[\Delta\hat{V}_\Delta]\|\boldsymbol{w}\|^2}{2} - \sqrt{d}\sum_k \hat{m}_k\boldsymbol{w}^\top\boldsymbol{\mu}_k - \sqrt{\mathbb{E}_\Delta[\Delta\hat{Q}_\Delta]}\boldsymbol{\xi}^\top\boldsymbol{w} + \lambda r(\boldsymbol{w})\right) \in \mathbb{R}^d \quad (43)$$

We can rewrite the prior contribution $\Psi_w$ as

$$\Psi_w(\hat{m},\hat{Q},\hat{V}) = -\frac{1}{d}\mathbb{E}_\xi\left[\frac{\mathbb{E}_\Delta[\Delta\hat{V}_\Delta]\|\boldsymbol{g}\|^2}{2} - \sqrt{d}\sum_k \hat{m}_k\boldsymbol{g}^\top\boldsymbol{\mu}_k - \sqrt{\mathbb{E}_\Delta[\Delta\hat{Q}_\Delta]}\boldsymbol{\xi}^\top\boldsymbol{g} + \lambda r(\boldsymbol{g})\right] \quad (44)$$

The analogy between the two contributions is evident, aside from the different dimensionality of the involved objects. The replica symmetric free energy (23) in the $\beta \to +\infty$ limit is computed by extremising with respect to the introduced order parameters,

$$f_{\text{RS}} = \underset{\substack{m,Q,V,b \\ \hat{m},\hat{Q},\hat{V}}}{\text{Extr}} \left[ \sum_{k=\pm} \hat{m}_k m_k + \frac{\mathbb{E}_\Delta \left[ \hat{V}_\Delta Q_\Delta - \hat{Q}_\Delta V_\Delta \right]}{2} - \alpha \Psi_{\text{out}}(m,Q,V) - \Psi_w(\hat{m},\hat{Q},\hat{V}) \right]. \tag{45}$$

To do so, we have to write down a set of saddle-point equations and solve them.

**Saddle-point equations**  The saddle-point equations are derived straightforwardly from the obtained free energy functionally extremising with respect to all parameters. A first set of equations is obtained from $\Psi_{\text{out}}$. Introducing

$$f_k \equiv \frac{h_k - \omega_k}{V_\Delta} \tag{46}$$

(and keeping in mind that this quantity is also $\Delta$-dependent) we have

$$\hat{Q}_\Delta = \alpha \mathbb{E}_{k,\zeta} \left[ f_k^2 \right], \quad \hat{V}_\Delta = -\frac{\alpha \mathbb{E}_{k,\zeta} \left[ f_k \zeta \right]}{\sqrt{Q_\Delta}}, \quad \hat{m}_k = \alpha \rho_k \mathbb{E}_{\Delta,\zeta} \left[ f_k \right], \quad b = \mathbb{E}_{k,\Delta,\zeta} \left[ h_k - m_k \right]. \tag{47}$$

Denoting $\hat{q} := \mathbb{E}[\Delta \hat{Q}_\Delta]$ and $\hat{v} := \mathbb{E}[\Delta \hat{V}_\Delta]$, the saddle-point equations from $\Psi_w$ are

$$V_\Delta = \frac{1}{d} \frac{\Delta \mathbb{E}_{\boldsymbol{\xi}} [\boldsymbol{g}^\top \boldsymbol{\xi}]}{\sqrt{\hat{q}}} \qquad Q_\Delta = \frac{\Delta}{d} \mathbb{E}_{\boldsymbol{\xi}} [\|\boldsymbol{g}\|^2] \qquad m_k = \frac{1}{\sqrt{d}} \mathbb{E}_{\boldsymbol{\xi}} \left[ \boldsymbol{g}^\top \boldsymbol{\mu}_k \right]. \tag{48}$$

Observe that the equations above imply that $V_\Delta = v\Delta$ and $Q_\Delta = q\Delta$, where $v$ and $q$ are some constant that do not depend on $\Delta^2$. We can rewrite

$$v = \frac{\mathbb{E}_{\boldsymbol{\xi}} [\boldsymbol{g}^\top \boldsymbol{\xi}]}{d\sqrt{\hat{q}}} \qquad q = \frac{\mathbb{E}_{\boldsymbol{\xi}} [\|\boldsymbol{g}\|^2]}{d} \qquad m_k = \frac{\mathbb{E}_{\boldsymbol{\xi}} \left[ \boldsymbol{g}^\top \boldsymbol{\mu}_k \right]}{\sqrt{d}} \tag{49}$$

and the remaining equations as

$$\hat{q} = \alpha \mathbb{E}_{k,\zeta,\Delta} \left[ \Delta f_k^2 \right], \quad \hat{v} = -\frac{\alpha}{\sqrt{q}} \mathbb{E}_{k,\Delta,\zeta} \left[ \sqrt{\Delta} f_k \zeta \right], \quad \hat{m}_k = \alpha \rho_k \mathbb{E}_{\Delta,\zeta} \left[ f_k \right], \quad b = \mathbb{E}_{k,\Delta,\zeta} \left[ h_k - m_k \right]. \tag{50}$$

To obtain the replica symmetric free energy, therefore, the given set of equation has to be solved, and the result then plugged in Eq. (45).

## A.2  Training and test errors

The order parameters introduced to solve the problem allow us to reach our ultimate goal of computing the average errors of the learning process. We will start with the estimation of the training loss

$$\epsilon_\ell \equiv \frac{1}{n} \sum_{\nu=1}^n \ell \left( y^\nu, \frac{\boldsymbol{w}^\star \boldsymbol{x}^\nu}{\sqrt{d}} + b^\star \right) \tag{51}$$

in the $n \to +\infty$ limit. The complication in computing this quantity is that the order parameters found in the learning process are, of course, correlated with the dataset $\mathcal{D}$ used for the learning itself. The best way to proceed is to observe that

$$\mathbb{E}_{\mathcal{D}} [\mathcal{R}(\boldsymbol{w}^\star, b^\star)] = - \lim_{\beta \to +\infty} \mathbb{E}_{\mathcal{D}} [\partial_\beta \ln \mathcal{Z}_\beta] = \lambda \mathbb{E}_{\mathcal{D}} [r(\boldsymbol{w}^\star)] + \epsilon_\ell$$

where

$$\epsilon_\ell = - \lim_{\beta \to +\infty} \partial_\beta (\beta \Psi_{\text{out}}) = \lim_{\beta \to +\infty} \mathbb{E}_{k,\Delta,\zeta} \left[ \int \frac{\ell_k(\eta) \, e^{-\frac{\beta(\eta - m_k^\star)^2}{2\Delta v^\star} - \beta \ell_k(\eta)}}{\sqrt{2\pi \beta^{-1} v^\star \Delta} \, Z_k(\omega_k^\star, \beta^{-1} v^\star \Delta)} \, d\eta \right]. \tag{52}$$

---

[2] This was largely expected in our setting, but we preferred to keep a redundant derivation as this factorisation cannot be performed when the derivation is generalised to the case of random covariance matrices which are not multiple of the identity.

In the $\beta \to +\infty$ limit, the integral concentrates on the minimiser of the exponent, that is, by definition, the proximal $h_k$. In conclusion,

$$\epsilon_\ell = \mathbb{E}_{k,\Delta,\xi}[\ell_k(h_k)]. \tag{53a}$$

By means of the same concentration result, the training error is

$$\epsilon_t = \frac{1}{n}\sum_{\nu=1}^n \mathbb{I}\left(\varphi\left(\frac{\boldsymbol{w}^{\star\mathsf{T}}\boldsymbol{x}^\nu}{\sqrt{d}}+b^\star\right)\neq y^\nu\right) \xrightarrow{n\to+\infty} \mathbb{E}_{k,\Delta,\zeta}\left[\mathbb{I}(\varphi(h_k)\neq y_k)\right]. \tag{53b}$$

The expressions above hold in general, but, as anticipated, important simplifications can occur in the set of saddle-point equations (50) and (49) depending on the choice of the loss $\ell$ and of the regularisation function $r$. The generalisation (or test) error can be written instead as

$$\epsilon_g = \mathbb{E}_{(y^{\text{new}},\boldsymbol{x}^{\text{new}})}\left[\mathbb{I}\left(\varphi\left(\frac{\boldsymbol{w}^{\star\mathsf{T}}\boldsymbol{x}^{\text{new}}}{\sqrt{d}}+b^\star\right)\neq y^{\text{new}}\right)\right]. \tag{53c}$$

This expression can be rewritten as

$$
\begin{aligned}
\epsilon_g &= \mathbb{E}_k\left[\int \mathbb{I}(\varphi(\eta)=y_k)\mathbb{E}_{\boldsymbol{x}^{\text{new}}}\left[\delta\left(\eta-\frac{\boldsymbol{w}^{\star\mathsf{T}}\boldsymbol{x}^{\text{new}}}{\sqrt{d}}-b^\star\right)\right]\mathrm{d}\eta\right]\\
&= \mathbb{E}_{k,\Delta}\left[\int \mathbb{I}(\varphi(\eta)=y_k)\mathbb{E}_{\boldsymbol{x}^{\text{new}}|\Delta}\left[\delta\left(\eta-\frac{\boldsymbol{w}^{\star\mathsf{T}}\boldsymbol{x}^{\text{new}}}{\sqrt{d}}-b^\star\right)\right]\mathrm{d}\eta\right]\\
&\xrightarrow{d\to+\infty} \mathbb{E}_{k,\Delta}\left[\int \mathbb{I}(\varphi(\eta)=y_k)\mathcal{N}(\eta|m_k^\star+b^\star,\Delta q^\star)\mathrm{d}\eta\right]\\
&= \mathbb{E}_{k,\Delta,\zeta}\left[\mathbb{I}\left(\varphi\left(m_k^\star+\sqrt{\Delta q^\star}\zeta+b^\star\right)\neq y_k\right)\right].
\end{aligned} \tag{53d}
$$

This can be easily computed numerically once the order parameters (including their functional dependence on $\Delta$) are given.

### A.3 Ridge regularisation

Let us fix now $r(\boldsymbol{w}) = \frac{1}{2}\|\boldsymbol{w}\|^2$. In this case, the computation of $\Psi_w$ can be performed explicitly via a Gaussian integration, and the saddle-point equations can take a more compact form that is particularly suitable for a numerical solution. In particular

$$\boldsymbol{g} = \frac{\sqrt{d}\sum_k \hat{m}_k\boldsymbol{\mu}_k + \sqrt{\hat{q}}\boldsymbol{\xi}}{\lambda+\hat{v}} \tag{54}$$

so that the prior saddle-point equations obtained from $\Psi_w$ become

$$v = \frac{1}{\lambda+\hat{v}} \tag{55a}$$

$$q = \frac{\sum_{kk'}\hat{m}_k\hat{m}_{k'}\boldsymbol{\mu}_k^\mathsf{T}\boldsymbol{\mu}_{k'}+\hat{q}}{(\lambda+\hat{v})^2} \tag{55b}$$

$$m_k = \frac{\sum_{k'}\hat{m}_{k'}\boldsymbol{\mu}_{k'}^\mathsf{T}\boldsymbol{\mu}_k}{\lambda+\hat{v}}. \tag{55c}$$

**Quadratic loss**  If we consider a quadratic loss $\ell(y,x) = \frac{1}{2}(y-x)^2$, then an explicit formula for the proximal can be found, namely

$$f_k = \frac{y_k-\omega_k}{1+v\Delta} \tag{56}$$

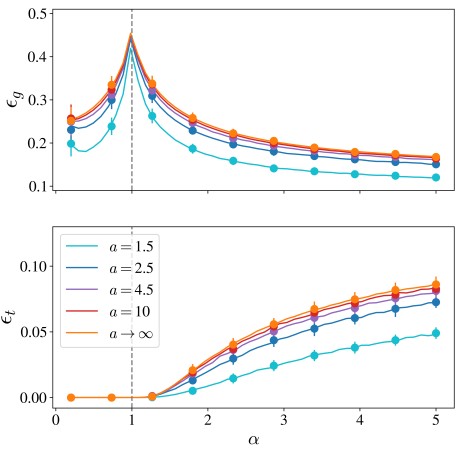

Figure 7: Test error $\epsilon_g$ (*top*) and training error $\epsilon_t$ (*bottom*) as predicted by Eq. (8) in the unbalanced $\rho = 1/4$ case. The dataset distribution is parametrised as in Eq. (12). The classification task is solved using a quadratic loss with ridge regularisation with $\lambda = 10^{-5}$. Dots correspond to the average outcome of 50 numerical experiments in dimension $d = 10^3$. In our parametrisation, the population covariance is $\boldsymbol{\Sigma} = \boldsymbol{I}_d$ for all values of $a$ and moreover, for $a \to +\infty$, the case of pure Gaussian clouds $P(\boldsymbol{x}|\boldsymbol{\mu}) = \mathcal{N}(\boldsymbol{x}|\boldsymbol{\mu},\boldsymbol{I}_d)$ is recovered.

so that the second set of saddle-point equations (50) can be written as

$$\hat{q} = \alpha \mathbb{E}_{k,\Delta} \left[ \frac{\Delta(y_k - m_k - b)^2}{(1 + v\Delta)^2} \right] + \alpha q \mathbb{E}_\Delta \left[ \frac{\Delta^2}{(1 + v\Delta)^2} \right], \tag{57a}$$

$$\hat{v} = \alpha \mathbb{E}_\Delta \left[ \frac{\Delta}{1 + v\Delta} \right] = \frac{1 - v\lambda}{v}, \tag{57b}$$

$$\hat{m}_k = \alpha \rho_k (y_k - m_k - b) \mathbb{E}_\Delta \left[ \frac{1}{1 + v\Delta} \right]. \tag{57c}$$

so that $v$ satisfies the self-consistent equation

$$\frac{1 - v\lambda}{v\alpha} = \mathbb{E}_\Delta \left[ \frac{\Delta}{1 + v\Delta} \right]. \tag{58}$$

As a complement to the information given in the main text, in Fig. 7 we give some numerical results for the case of *unbalanced* clusters, which show a perfect agreement between the theoretical predictions and the numeral results.

**Logistic loss**  Let us now consider $K = 2$ and $\mathcal{Y} = \{-1, 1\}$: in the following, we will label the different classes by $k = \pm$. In this context, we discuss the relevant case of the logistic loss $\ell_\pm(x) = \ln(1 + e^{\mp x})$. The proximal equation for this loss is the solution of the equations:

$$f_\pm = \arg \min_u \left[ \frac{\Delta v u^2}{2} + \ln\left(1 + e^{\mp(\Delta v u + \omega_\pm)}\right) \right] \tag{59}$$

having only one solution for which, however, there is no closed-form expression; the equation can be solved numerically. Interestingly, the $\lambda \to 0$ limit of such loss recovers the hinge loss with zero margin [64]. It is numerically convenient in this case to consider $\hat{v} \mapsto \lambda\hat{v}$, $v \mapsto \lambda^{-1}v$, $m \mapsto \lambda^{-1}m$, $b \to \lambda^{-1}b$ and $q \mapsto \lambda^{-2}q$: this provides a new set of consistent equations for the rescaled variables, implying that in the $\lambda \to 0$ limit the original order parameters diverge.

The zero-regularisation limit of the logistic loss can help us study the separability transition. To obtain the position of the separability transition, we follow the derivation proposed by Mignacco et al. [49]. Let us assume for simplicity that $\mu_+ = -\mu_- \equiv \mu$ with $\|\mu\| = 1$ for simplicity. It is immediate to see that in this case $m_+ = -m_- \equiv m$, and it is, therefore, possible to introduce $\hat{m} := \hat{m}_+ + \hat{m}_-$, so that the saddle-point equations can be written as

$$\begin{cases} \hat{q} = \alpha \mathbb{E}_{\pm,\zeta,\Delta} \left[ \Delta f_\pm^2 \right], \\ \hat{v} = -\frac{\alpha}{\sqrt{q}} \mathbb{E}_{\pm,\Delta,\zeta} \left[ \sqrt{\Delta} f_\pm \zeta \right], \\ \hat{m} = \alpha \mathbb{E}_{\pm,\Delta,\zeta} \left[ f_\pm \right], \\ b = \mathbb{E}_{\pm,\Delta,\zeta} \left[ h_\pm \mp m \right], \end{cases} \qquad \begin{cases} v = \frac{1}{\lambda + \hat{v}} \\ q = \frac{\hat{m}^2 + \hat{q}}{(\lambda + \hat{v})^2} \\ m = \frac{\hat{m}}{\lambda + \hat{v}}. \end{cases} \tag{60}$$

We can start with the fact that

$$\alpha v^2 \mathbb{E}_{\pm,\zeta,\Delta} [\Delta f_\pm^2] = v^2 \hat{q} = q - m^2. \tag{61}$$

Introducing $\theta = \frac{m}{\sqrt{q}}$, $\tilde{b} = \frac{b}{\sqrt{q}}$ and $\tilde{v} = \frac{v}{\sqrt{q}}$, then we can re-write the equation as

$$\alpha \mathcal{S}(\tilde{v}, \theta, \tilde{b}, q) = 1 - \theta^2, \qquad \mathcal{S}(\tilde{v}, \theta, \tilde{b}, q) := \tilde{v}^2 \mathbb{E}_{\pm,\zeta,\Delta} [\Delta f_\pm^2]. \tag{62}$$

We state now that the separable phase corresponds to the limit $\tilde{v} \to +\infty$ as $\lambda \to 0$ when using the logistic loss, keeping $\theta$ and $b$ fixed. In this case, indeed, we have that $f_\pm = \ell'_\pm(h_\pm) \to 0$, i.e., $h_\pm \to \pm +\infty$, which is the condition to have separability, $\lim_{\lambda \to 0} \epsilon_\ell = \lim_{\lambda \to 0} \mathbb{E}[\ell_\pm(h_\pm)] = 0$.

To properly take this limit we need an explicit expression of $\mathcal{S}(\theta, \tilde{b}, q)$, we observe that $h_\pm$ satisfies $h_\pm + v\Delta\ell'_\pm(h_\pm) = \omega_\pm$, with $\omega_\pm := \pm m + b + \sqrt{q\Delta}\zeta$ and $f_\pm := \ell'_\pm(h_\pm)$: it is possible therefore introduce the Legendre function $\tilde{\ell}_\pm(f) = \max_h \{hf - \ell_\pm(h)\}$ of $\ell_\pm(h)$, such that $v\Delta f_\pm + \tilde{\ell}'_\pm(f_\pm) = \omega_\pm$. If now we want to compute the probability that $f_\pm < f$ at fixed label and value of $\Delta$, we have

$$\mathbb{P}[f_\pm \leq f | \pm, \Delta] = \mathbb{P}[\pm m + b + \sqrt{q\Delta}\zeta \leq v\Delta f + \tilde{\ell}'(f) | \pm, \Delta]$$

$$= \Phi\left( \frac{v\Delta f \mp m - b + \tilde{\ell}'_\pm(f)}{\sqrt{q\Delta}} \right), \tag{63}$$

with $\Phi(z) = \mathbb{P}[\zeta \le z]$ for a standard Gaussian random variable $z \sim \mathcal{N}(0, 1)$. By consequence, using the fact that $\mathbb{E}_\zeta[f_\pm^2] = -2\int_{-1}^0 f\mathbb{P}[f_\pm \le f|\pm, \Delta]df$, and observing that $-1 \le f_\pm \le 0$

$$
\begin{aligned}
\mathcal{S}(\tilde{v},\theta,\tilde{b},q) &:= 2\tilde{v}^2 \int_0^1 f\mathbb{E}_\Delta\left[\Delta\rho_+\Phi\left(\frac{-\tilde{v}\Delta f-\theta-\tilde{b}}{\sqrt{\Delta}}+\frac{\tilde{\ell}'_+(-f)}{\sqrt{2q\Delta}}\right)+\Delta\rho_-\Phi\left(\frac{-\Delta\tilde{v}f+\theta-\tilde{b}}{\sqrt{\Delta}}+\frac{\tilde{\ell}'_-(-f)}{\sqrt{2q\Delta}}\right)\right]d \\
&= 2\int_0^{\tilde{v}} f\mathbb{E}_\Delta\left[\Delta\rho_+\Phi\left(\frac{-\Delta f-\theta-\tilde{b}}{\sqrt{\Delta}}+\frac{\tilde{\ell}'_+(-f/\tilde{v})}{\sqrt{2q\Delta}}\right)+\Delta\rho_-\Phi\left(\frac{-\Delta f+\theta-\tilde{b}}{\sqrt{\Delta}}+\frac{\tilde{\ell}'_-(-f/\tilde{v})}{\sqrt{2q\Delta}}\right)\right]df.
\end{aligned}
\tag{64}
$$

Eq. (62) allows us to express $\tilde{v} \equiv \tilde{v}(\theta, b, q)$. It turns out that $\tilde{v} = \tilde{v}(\theta, \tilde{b}, q)$ is finite for any finite $q$, and, at $\tilde{b}$ and $\theta$ fixed, can diverge for $q \to +\infty$ only. If we assume $q$ to be finite and we take $\lim_{\tilde{v}\to+\infty} \mathcal{S}(\tilde{v}, \theta, \tilde{b}, q)$, the expression diverges giving therefore an inconsistency (as $\tilde{\ell}'_\pm(-f/\tilde{v}) \to \pm\infty$). On the other hand, by taking the limit $q \to +\infty$, the function $\lim_{q\to+\infty} \mathcal{S}(\tilde{v}, \theta, \tilde{b}, q)$ has a finite limit and it is monotonically increasing in $\tilde{v}$, so that Eq. (62) allows for a finite $\tilde{v}$ as long as

$$
\alpha > \frac{1-\theta^2}{\mathcal{S}_\star(\theta, \tilde{b})},
\tag{65}
$$

where

$$
\begin{aligned}
\mathcal{S}_\star(\theta,\tilde{b}) &= \lim_{\tilde{v}\to+\infty}\lim_{q\to+\infty}\mathcal{S}(\tilde{v},\theta,\tilde{b},q) \\
&= 2\int_0^\infty f\mathbb{E}_\Delta\left[\Delta\rho_+\Phi\left(\frac{-\Delta f-\theta-\tilde{b}}{\sqrt{\Delta}}\right)+\Delta\rho_-\Phi\left(\frac{-\Delta f+\theta-\tilde{b}}{\sqrt{\Delta}}\right)\right]df \\
&= \int_0^\infty f^2\mathbb{E}_\Delta\left[\rho_+\mathcal{N}\left(f+\frac{\theta+\tilde{b}}{\sqrt{\Delta}}\Big|0,1\right)+\rho_-\mathcal{N}\left(f+\frac{\theta-\tilde{b}}{\sqrt{\Delta}}\Big|0,1\right)\right]df.
\end{aligned}
\tag{66}
$$

As a result, given that $\theta \in (0, 1]$, the smaller value for which $\tilde{v}$ is finite is

$$
\alpha^\star = \max_{\theta\in(0,1],b}\frac{1-\theta^2}{\mathcal{S}_\star(\theta,\tilde{b})}.
\tag{67}
$$

This corresponds to the threshold value for the separability transition.

### A.4 The Bayes optimal error in the $K = 2$ case

We derive here the Bayes optimal error in the case of two clusters $K = 2$ with centroids in $1/\sqrt{d}\boldsymbol{\mu}_\pm = \pm1/\sqrt{d}\boldsymbol{\mu}$, with $\boldsymbol{\mu} \sim \mathcal{N}(\mathbf{0}, \boldsymbol{I}_d)$. The derivation is a variation of the arguments in Ref. [49]. Given an estimate of $\boldsymbol{\mu}$, it is possible to compute $p(y, \boldsymbol{x}|\boldsymbol{\mu}) = p(\boldsymbol{x}|y, \boldsymbol{\mu})p(y) = \mathbb{E}[\mathcal{N}(\boldsymbol{x}; y/\sqrt{d}\boldsymbol{\mu}, \Delta\boldsymbol{I})]p(y)$ with $p(y) = \rho\delta_{y,+1} + (1-\rho)\delta_{y,-1}$. We assume for now that both $\mathbb{E}[\Delta]$ and $\mathbb{E}[\Delta^{-2}]$ are finite. We can write down the posterior for $\boldsymbol{\mu}$ given a dataset $\mathcal{D} = \{(y_\nu, \boldsymbol{x}_\nu)\}_{\nu=1}^n$ of observations as

$$
p(\boldsymbol{\mu}|\mathcal{D}) \propto p(\mathcal{D}|\boldsymbol{\mu})p(\boldsymbol{\mu}) \propto \mathbb{E}\left[\frac{\exp\left(-\frac{\|\boldsymbol{\mu}\|^2}{2}-\sum_{\nu=0}^n\frac{\|\boldsymbol{x}_\nu-y_\nu\boldsymbol{\mu}\|^2}{2\Delta_\nu}\right)}{\prod_\nu(2\pi\Delta_\nu)^{d/2}}\right]
\tag{68}
$$

where the expectation is over the set $\boldsymbol{\Delta} := \{\Delta_\nu\}_{\nu\in[n]}$. Given a new pair $(y_0, \boldsymbol{x}_0)$, we can now estimate

$$
\mathbb{P}[y_0 = \pm1|(\boldsymbol{x}_0, \Delta_0), \{(y_\nu, \boldsymbol{x}_\nu, \Delta_\nu)\}_{\nu=1}^n] \propto p(y_0)\int d\boldsymbol{\mu}\,\frac{e^{-\frac{\|\boldsymbol{\mu}\|^2}{2}-\sum_{\nu=0}^n\frac{1}{2\Delta_\nu}\left\|\boldsymbol{x}_\nu-\frac{y_\nu\boldsymbol{\mu}}{\sqrt{d}}\right\|^2}}{\prod_\nu(2\pi\Delta_\nu)^{d/2}},
\tag{69}
$$

where we condition on the values of the set $\{\Delta_\nu\}_{\nu=0}^n$ as well. The integral can be computed as

$$
\int d\boldsymbol{\mu}\,e^{-\frac{\|\boldsymbol{\mu}\|^2}{2}-\sum_{\nu=0}^n\frac{1}{2\Delta_\nu}\left\|\boldsymbol{x}_\nu-\frac{y_\nu\boldsymbol{\mu}}{\sqrt{d}}\right\|^2} \propto \frac{1}{(1+\alpha/\delta_n)^{d/2}}\exp\left(\frac{\alpha y_0\boldsymbol{x}_0}{1/d+\Delta_0(1+\alpha/\delta_n)}\frac{1}{n}\sum_{\nu=1}^n\frac{y_\nu\boldsymbol{x}_\nu}{\Delta_\nu}\right)
\tag{70}
$$

where we have omitted a coefficient independent of $y_0$, and we have introduced the harmonic mean

$$
\delta_n := \left(\frac{1}{n}\sum_{\nu=1}^n\frac{1}{\Delta_\nu}\right)^{-1}.
\tag{71}
$$

For the sake of brevity, let us an auxiliary random variable correlated with the harmonic mean, namely

$$\hat{\delta}_n := \left( \frac{1}{n} \sum_{\nu=1}^{n} \frac{\Delta'_\nu}{\Delta_\nu^2} \right)^{-1}$$

where $\{\Delta'_\nu\}_{\nu \in [n]}$ is a distinct and independent set of values of $\Delta$'s. Now we have that, by writing for $\nu \in [n]$ $\mathbf{x}_\nu = y_\nu \boldsymbol{\mu} + \sqrt{\Delta'_\nu} \boldsymbol{\zeta}_\nu$, $\boldsymbol{\zeta}_\nu$ vector distributed as $\mathcal{N}(\mathbf{0}, \boldsymbol{I}_d)$, and $\mathbf{x}_0 = y_0^\star \boldsymbol{\mu} + \sqrt{\Delta_0^\star} \boldsymbol{\zeta}_0$, $\boldsymbol{\zeta}_0$ also distributed as $\mathcal{N}(\mathbf{0}, \boldsymbol{I}_d)$, then up to $O(1/d)$ contributions

$$
\begin{aligned}
\frac{1}{n} \sum_{\nu=1}^{n} \frac{y_\nu \mathbf{x}_0^\top \mathbf{x}_\nu}{\Delta_\nu} &= \frac{y_0^\star}{\delta_n} + \frac{y_0^\star}{n} \sum_{\nu=1}^{n} \frac{y_\nu \sqrt{\Delta'_\nu} \boldsymbol{\mu}^\top \boldsymbol{\zeta}_\nu}{\Delta_\nu} + \frac{\sqrt{\Delta_0^\star} \boldsymbol{\mu}^\top \boldsymbol{\zeta}_0}{\delta_n} + \frac{1}{n} \sum_{\nu=1}^{n} \frac{y_\nu \sqrt{\Delta'_\nu \Delta_0^\star} \boldsymbol{\zeta}_0^\top \boldsymbol{\zeta}_\nu}{\Delta_\nu} + O(1/d) \\
&\stackrel{\mathrm{d}}{=} \frac{y_0^\star}{\delta_n} + \sqrt{\frac{\Delta_0^\star}{\delta_n^2} + \frac{\Delta_0^\star}{\alpha \hat{\delta}_n}} \zeta + O(1/d) = \frac{y_0^\star + \sqrt{\Delta_0^\star B(\delta_n, \hat{\delta}_n)}}{\delta_n} + O(1/d),
\end{aligned}
\tag{72}
$$

at the leading order in $n, d$, where $\zeta \sim \mathcal{N}(0, 1)$. We have also introduced $B(\delta, \hat{\delta}) := 1 + \frac{\delta^2}{\alpha \hat{\delta}}$. In the large $n, d$ limit therefore

$$\mathbb{P}[y_0 = \pm 1 | (\mathbf{x}_0, \Delta_0), \{(y_\nu, \mathbf{x}_\nu, \Delta_\nu)\}_{\nu=1}^{n}] \propto \exp\left[ \frac{y_0 A(\delta_n)}{\Delta_0} \left( y_0^\star + \sqrt{\Delta_0^\star B(\delta_n, \hat{\delta}_n)} \zeta \right) + \ln p(y_0) \right] \tag{73}$$

where $A(\delta) := \frac{\alpha}{\alpha + \delta}$. The conditional optimal estimator is then

$$\hat{y}_0 |_{\Delta_0, \Delta} = \arg \max_{y_0 \in \{-1, 1\}} \left[ \frac{y_0 A(\delta_n)}{\Delta_0} \left( y_0^\star + \sqrt{\Delta_0^\star B(\delta_n, \hat{\delta}_n)} \zeta \right) + \ln p(y_0) \right] \tag{74}$$

the dependence on $\mathbf{x}^0$ being expressed by $\zeta$ and $\Delta_0^\star$. The probability that such an estimator is in fact not exact is

$$\mathbb{P}[\hat{y}_0 \neq y_0^\star | \Delta_0, \Delta] = \mathbb{P}\left[ y_0^\star \zeta < -\frac{1}{\sqrt{\Delta_0^\star B(\delta_n, \hat{\delta}_n)}} \left( 1 + \frac{\Delta_0}{2A(\delta_n)} \ln \frac{p(y_0^\star)}{p(-y_0^\star)} \right) \right]. \tag{75}$$

If $\hat{\Phi}(x) := 1 - \Phi(x) = \frac{1}{\sqrt{2\pi}} \int_x^\infty e^{-t^2/2} \, \mathrm{d}t$, then the Bayes optimal error is therefore

$$\varepsilon_g^{\mathrm{BO}} = \rho \mathbb{E}\left[ \hat{\Phi}(\kappa_+) \right] + (1 - \rho) \mathbb{E}\left[ \hat{\Phi}(\kappa_-) \right], \tag{76}$$

where, observing that $\delta_n^{-1} \to \mathbb{E}[\Delta^{-1}]$ and $\hat{\delta}_n^{-1} \to \mathbb{E}[\Delta] \mathbb{E}[\Delta^{-2}]$,

$$\kappa_\pm \equiv \kappa_\pm(\Delta_0, \Delta_0^\star) := \frac{1 \pm \frac{\Delta_0}{2} \left( 1 + \frac{1}{\alpha \mathbb{E}[\Delta^{-1}]} \right) \ln \frac{\rho}{1 - \rho}}{\sqrt{\Delta_0^\star \left( 1 + \frac{\mathbb{E}[\Delta] \mathbb{E}[\Delta^{-2}]}{\alpha \mathbb{E}[\Delta^{-1}]^2} \right)}}.$$

## A.5 The uncorrelated-teacher case and universality in binary classification

In the present section, we will focus on the $K = 2$ case, assuming labels to be given by $\mathcal{Y} = \{-1, +1\}$. In a recent paper, Gerace et al. [25] showed that a classification task on Gaussian clouds exhibits universality features in the case in which the labels are randomly assigned to the dataset points. Under the hypothesis of a loss function satisfying $\ell(y, \eta) = \ell(-y, -\eta)$ with ridge regularisation, they show that a dataset obtained from a mixture of Gaussian clouds with equal covariance $\boldsymbol{\Sigma}$ is equivalent to a dataset obtained from a single Gaussian cloud with zero mean and covariance $\boldsymbol{\Sigma}$. Moreover, using ridge regression, the training loss $\epsilon_\ell$ is shown to depend on the sample complexity only (and not on $\boldsymbol{\Sigma}$) in the $\lambda \to 0^+$ limit. This result has been generalised immediately afterwards by Pesce et al. [59], who showed that the same picture holds in the case in which labels are generated by a "teacher" modeled by a distribution $P_0(y|\tau)$, with $P_0(y|\tau) = P_0(-y| - \tau)$, and parametrised by vector $\boldsymbol{\theta}_0 \in \mathbb{R}^d$, which is *uncorrelated* with the data structure.

In our setting, this would amount to considering a database $\mathcal{D}$ generated from the joint distribution

$$P(\boldsymbol{x}, y) = P_0\left(y \middle| \frac{\boldsymbol{\theta}_0^\top \boldsymbol{x}}{\sqrt{d}}\right) \sum_{k=\pm} \rho_k \mathbb{E}[\mathcal{N}(\boldsymbol{x}|\boldsymbol{\mu}_k, \Delta \boldsymbol{I}_d)], \tag{77}$$

the random labels case corresponding to $P_0(y|\tau)$ given by a Rademacher distribution in $y$ independent of $\tau$. The condition of uncorrelated teacher is expressed by[3]

$$\lim_{d\to+\infty} \frac{\boldsymbol{\theta}_0^\top \boldsymbol{\mu}_\pm}{d} = 0. \tag{78}$$

Let us assume for simplicity that we adopt ridge regularisation, $r(\boldsymbol{w}) = \frac{\lambda}{2}\|\boldsymbol{w}\|^2$. The analysis of this setting is perfectly analogous to the one discussed above, but provides slightly different fixed-point equations for the order parameters, and in particular, it requires introducing the overlap between the weights $\boldsymbol{w}$ and the teacher parameter $\boldsymbol{\theta}_0$, $t = \frac{1}{d}\boldsymbol{w}^\top \boldsymbol{\theta}_0$, and its corresponding Lagrange multiplier $\hat{t}$. Following therefore a procedure that combines the one presented above and the derivation given in Ref. [59] for the Gaussian case, we can obtain the following fixed-point equations,

$$
\begin{cases}
v = \frac{1}{\lambda + \hat{v}} \\
q = \frac{\gamma \hat{t}^2 + \|\sum_c \hat{m}_c \boldsymbol{\mu}_c\|^2 + q}{(\lambda + \hat{v})^2}, \\
m_\pm = \frac{\sum_{c'} \hat{m}_{c'} \boldsymbol{\mu}_{c'}^\top \boldsymbol{\mu}_\pm}{\lambda + \hat{v}}, \\
t = \frac{\gamma \hat{t}}{\lambda + \hat{v}}
\end{cases}
\qquad
\begin{cases}
\hat{q} = \alpha \sum_y \mathbb{E}_{\pm, \Delta, \zeta}\left[\Delta Z_0\left(y, \sqrt{\frac{\Delta}{q}}t\zeta, \Delta\gamma - \Delta\frac{t^2}{q}\right) f^2(y, \omega_\pm, v)\right], \\
\hat{v} = -\alpha \sum_y \mathbb{E}_{\pm, \Delta, \zeta}\left[Z_0\left(y, \sqrt{\frac{\Delta}{q}}t\zeta, \Delta\gamma - \Delta\frac{t^2}{q}\right) \partial_\omega f(y, \omega_\pm, v)\right], \\
\hat{m}_\pm = \alpha\rho_\pm \sum_y \mathbb{E}_{\Delta, \zeta}\left[Z_0\left(y, \sqrt{\frac{\Delta}{q}}t\zeta, \Delta\gamma - \Delta\frac{t^2}{q}\right) f_\pm(y, \omega_\pm, v)\right] \\
\hat{t} = \alpha \sum_y \mathbb{E}_{\pm, \Delta, \zeta}\left[\partial_\omega Z_0\left(y, \sqrt{\frac{\Delta}{q}}t\zeta, \Delta\gamma - \Delta\frac{t^2}{q}\right) \Delta f(y, \omega_\pm, v)\right],
\end{cases}
\tag{79}
$$

with $\gamma := \frac{1}{d}\|\boldsymbol{\theta}_0\|^2$ and

$$f(y, \omega, v) := \arg\min_u \left[\frac{\Delta v u^2}{2} + \ell(y, \omega + v\Delta u)\right], \quad \omega_\pm := b + m_\pm + \sqrt{\Delta q}\zeta \tag{80}$$

$$b = \sum_y \mathbb{E}_{\pm, \Delta, \zeta}\left[Z_0\left(y, \sqrt{\frac{\Delta}{q}}t\zeta, \Delta\rho - \Delta\frac{t^2}{q}\right)(\omega_\pm + v\Delta f(y, \omega_\pm, v) - m_\pm)\right].$$

that provide us the quantities to plug into Eqs. (53) to compute the asymptotic errors. In the equations above, $Z_0(y, \omega, v) := \mathbb{E}_z[P_0(y|\omega + \sqrt{v}z)]$ with $z \sim \mathcal{N}(0, 1)$.

**Mean universality** Following Ref. [59], let us now make the additional assumption $P_0(y|\tau) = P_0(-y|-\tau)$. This implies that $Z_0(y, \omega, v) = Z_0(-y, -\omega, v)$. In this case, we claim that the solution has $m_\pm = \hat{m}_\pm = b = 0$. It is clear that if $\hat{m}_\pm = 0$, then $m_\pm = 0$ and, moreover, because of parity, the solution $b = 0$ is consistent. On the other hand, if $m_\pm = 0$ and $b = 0$, remembering that $\ell(y, \eta) = \ell(-y, -\eta)$, then $f(1, \omega, v) = -f(-1, -\omega, v)$ so that the formula for $\hat{m}_\pm$ involves a Gaussian integral of an odd function which is, therefore, zero, implying $\hat{m}_\pm = 0$. We are left therefore with a much simpler set of fixed-point equations, namely

$$
\begin{cases}
v = \frac{1}{\lambda + \hat{v}} \\
q = \frac{\gamma \hat{t}^2 + q}{(\lambda + \hat{v})^2}, \\
t = \frac{\gamma \hat{t}}{\lambda + \hat{v}}
\end{cases}
\qquad
\begin{cases}
\hat{q} = \alpha \sum_y \mathbb{E}_{\Delta, \zeta}\left[\Delta Z_0\left(y, \sqrt{\frac{\Delta}{q}}t\zeta, \Delta\gamma - \Delta\frac{t^2}{q}\right) f^2(y, \omega, v)\right], \\
\hat{v} = -\alpha \sum_y \mathbb{E}_{\Delta, \zeta}\left[Z_0\left(y, \sqrt{\frac{\Delta}{q}}t\zeta, \Delta\gamma - \Delta\frac{t^2}{q}\right) \partial_\omega f(y, \omega, v)\right], \\
\hat{t} = \alpha \sum_y \mathbb{E}_{\Delta, \zeta}\left[\partial_\omega Z_0\left(y, \sqrt{\frac{\Delta}{q}}t\zeta, \Delta\gamma - \Delta\frac{t^2}{q}\right) \Delta f(y, \omega, v)\right],
\end{cases}
\tag{81}
$$

with

$$f(y, \omega, v) := \arg\min_u \left[\frac{\Delta v u^2}{2} + \ell(y, \omega + v\Delta u)\right], \quad \omega := \sqrt{\Delta q}\zeta, \tag{82}$$

which is exactly the formula we would have obtained assuming $\boldsymbol{\mu}_\pm = \boldsymbol{0}$, i.e., the presence of *one cloud only*, so that $P(\boldsymbol{x}, y) = P_0\left(y \middle| \frac{1}{\sqrt{d}}\boldsymbol{\theta}_0^\top \boldsymbol{x}\right)\mathbb{E}[\mathcal{N}(\boldsymbol{x}|\boldsymbol{0}, \Delta\boldsymbol{I}_d)]$. This *mean universality* result generalises the result in Ref. [59] for Gaussian clouds.

---

[3] In our case, this condition is simpler than in Ref. [59] as we assume that each class has the same homogeneous covariance.

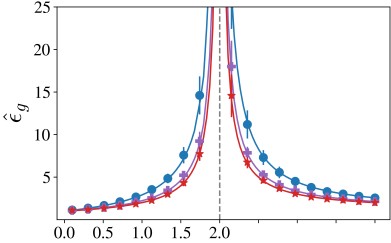 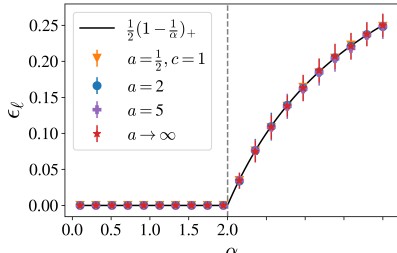

Figure 8: Test error $\hat{\epsilon}_g$ (**left**) and training loss $\epsilon_\ell$ (**right**) obtained by running numerical experiments for a classification task on *two* clouds with opposite means and randomly labeled points. Each cloud is generated with distribution in Eq. (11b) for different values of $a$ (i.e., different power-law decay). All clouds with $a > 1$ have the same covariance $\Sigma = I_d$ and are obtained using Eq. (12). The case $a = 1/2$ and $c = 1$, instead, corresponds to infinite $\sigma^2$ (note that in this case $\hat{\epsilon}_g$ is infinite and is therefore not plotted). For each $a$, the numerical experiments are compared with the theoretical prediction for a random label classification task on a single cloud centered in the origin and with the same parameter $a$, showing an excellent agreement. Note that the training loss is found to be universal and following (87), independently from the variance distribution.

**Random labels under square loss**   The case of *random label* is particularly interesting as it exhibits further universality when the square loss is adopted. In this case, $P_0(y|\tau)$ is simply the Rademacher distribution. Using $\ell(y, \eta) = \frac{1}{2}(y - \eta)^2$, the equations greatly simplify and we obtain

$$\begin{cases} v = \frac{1}{\lambda + \hat{v}}, \\ q = \frac{\hat{q}}{(\lambda + \hat{v})^2}, \end{cases} \quad \begin{cases} \hat{q} = \alpha \mathbb{E}_\Delta\left[\frac{\Delta + q\Delta^2}{(1 + v\Delta)^2}\right], \\ \hat{v} = \alpha \mathbb{E}_\Delta\left[\frac{\Delta}{1 + v\Delta}\right], \end{cases} \tag{83}$$

whereas $t = \hat{t} = 0$, so that the test error, obtained by computing $\varphi(y, \eta) = (y - \eta)^2$ where $\eta$ are taken to be pre-activations (because using post-activations would trivially yield $\epsilon_g = 1/2$ in this random label setting), and the training loss are

$$\hat{\epsilon}_g := \mathbb{E}_{(y, x)}\left[\left(y - \frac{1}{\sqrt{d}}x^\top w^\star\right)^2\right] = 1 + \sigma^2 q, \qquad \epsilon_\ell = \frac{1}{2}\mathbb{E}_\Delta\left[\frac{1 + q\Delta}{(1 + v\Delta)^2}\right]. \tag{84}$$

Note that the test error is infinite if $\sigma^2 = +\infty$. Introducing the notation $\delta_k := \mathbb{E}[(1 + v\Delta)^{-k}]$, the fixed-point equations above read

$$\begin{cases} v = \frac{1}{\lambda + \hat{v}}, \\ q = \frac{\hat{q}}{(\lambda + \hat{v})^2}, \end{cases} \quad \begin{cases} \hat{q} = \alpha\frac{\delta_1 - \delta_2}{v} + \alpha q \frac{1 - 2\delta_1 + \delta_2}{v^2}, \\ \hat{v} = \alpha\frac{1 - \delta_1}{v}, \end{cases} \quad , \quad \epsilon_\ell = \frac{1}{2}\left(\delta_2 + q\frac{\delta_1 - \delta_2}{v}\right). \tag{85}$$

Observe now that in the limit $\lambda \to 0$

$$x := \frac{q}{v} = \frac{\hat{q}}{\hat{v}} = -\frac{\delta_2 - 2\delta_1 + 1}{\delta_1 - 1}x + \frac{\delta_2 - \delta_1}{\delta_1 - 1}, \tag{86}$$

which is solved by $x = 1$, so that in this limit $\epsilon_\ell = \frac{1}{2}\delta_1$. But, on the other hand, the fixed point equation for $\hat{v}$ implies that $v$ is such that $\delta_1 = 1 - 1/\alpha$ if $\alpha \geq 1$, and zero otherwise (as $\delta_1 \geq 0$ by definition) so that we recover the universal formula for the training loss obtained by Gerace et al. [25],

$$\epsilon_\ell = \frac{1}{2}\left(1 - \frac{1}{\alpha}\right)_+, \tag{87}$$

where $(x)_+ = x\theta(x)$. Note that this formula *does not depend on the choice of the distribution of* $\Delta$. We verify this result in Fig. 8. We run numerical experiments using a quadratic loss on datapoints split in two clouds of equal weights centered around $\mu_1 = -\mu_2 \sim \mathcal{N}(0, I_d)$, and generated with distribution as in (12), parametrised by $a$, so that each cloud has $\Sigma = I_d$. Labels are assigned randomly with Rademacher distribution. The results are compared with the prediction for *one* cloud with $\mu = 0$ but with the same parameter $a$. We see that mean-independence in this setting is indeed verified.

# B    Note on numerical results

**State evolution equations**   Each average appearing in the update of the order parameters was performed using either the quadratic integration function `quad` from the `SciPy` package [73] or, typically in the case of infinite variance distributions, Monte-Carlo averaging was used with $10^5 - 10^6$ samples of the variance. For the convergence criterion, we use a tolerance of $10^{-5}$; the algorithm stops and returns the parameters after at most $10^3$ updates.

**Numerical experiments**   Numerical experiments regarding the quadratic loss with ridge regularisation were performed by computing the Moore-Penrose pseudoinverse solution. For the logistic loss, we used the `LogisticRegression` module from the `Scikit-learn` package [58].

**Code**   An implementation of the solution of the fixed-point equations and numerical simulations for the square loss and ridge regularisation can be found in https://github.com/urteado/super_classification.