# OpenReview forum: "Classification of Heavy-tailed Features in High Dimensions: a Superstatistical Approach"
_NeurIPS.cc/2023/Conference — NeurIPS 2023 poster_

### Official Review · Reviewer_7S1w · 2023-07-03

**Soundness:** 3 good
**Presentation:** 3 good
**Contribution:** 3 good
**Rating:** 7
**Confidence:** 3

**Summary:**

The paper is concerned with binary classification, when the data comes from two point clouds that are superposition of Gaussian distribution. This model allows for data distribution with fat tails. The authors analyse the performance of empirical risk minimization in the high-dimensional regime where the number of training samples and the dimension jointly diverge. Using the replica method from statistical physics, they reduce the computation of e.g the training loss / generalization error to the resolution of self-consistent equations.
In the third section of the paper, the authors apply their main result on experiments with synthetic data.

**Strengths:**

The paper is clearly written, and the mathematical derivations are easy to follow. The application of the replica method on a mixture of superposition of Gaussian distributions is new, to the best of my knowledge.

**Weaknesses:**

The paper lacks an experiment on real data to showcase situations in which the data model used in this paper (superposition of Gaussian) is more realistic / useful than simply using a mixture of two Gaussians.

**Questions:**

Could the computations be easily extended to  :
1) other types of estimators, e.g. Bayesian estimators that sample from a posterior distribution instead of ERM
2) more generic covariance for the Gaussian distributions. For instance instead of $N(\mu, \Delta I_d)$, use $N(\mu, \Delta \Sigma)$ where $\Sigma is a generic covariance matrix.

---

> ### Author Rebuttal · Authors · 2023-08-08
>
> We thank the referee for her/his positive comments on our paper. We summarise our answers to her/his questions below.
> * As foreseen by the referee, the method can indeed be applied to the study of an estimator obtained by minimisation of a proper convex function, and in particular, to the study of the Bayes optimal estimator itself (in this case, the convexity of the problem in the Bayes optimal setting is guaranteed by the so-called "Nishimori conditions" in the statistical physics jargon). As the Bayes optimal estimator is indeed an important reference for our results, stimulated by the question of the referee we have derived the expression for the Bayes optimal error in a new Appendix A.4 and added, as guide for comparison, the corresponding error curves in our plots.
> * The case suggested by the referee is indeed within the reach of our theory, which can be easily generalised to consider clouds of the type $P(\boldsymbol{x}|\boldsymbol \mu)=\mathbb E[\mathcal N(\boldsymbol x;\boldsymbol\mu,\Delta\boldsymbol\Sigma)]$, possibly with a different covariance matrix for each cloud (the case of identical $\boldsymbol\Sigma$ can be reduced to the current setting by a simple change of variables). For example, by supposing that the two clouds are described by a density $P_\pm(\boldsymbol x|\boldsymbol\mu_\pm)=\mathbb E[\mathcal N(\boldsymbol x;\boldsymbol\mu_\pm,\Delta\boldsymbol\Sigma)]$, fixed-point equations similar to the ones presented in the paper would be obtained with respect to scalar order parameters, although two order parameters $\boldsymbol q_\pm$ would be needed in this case, each taking into account the correlation between the estimator $\boldsymbol w^\star$ and each one of the covariances. To avoid this complication, we restricted our analysis to the simplest case for illustrative purposes: we however added a line in the main text commenting about this more general setting pointed out by the referee. We mention here that the case in which the matrix $\boldsymbol\Sigma$ is assumed to be random is, instead, much more challenging to analyse: the dependence of the order parameters on the additional stochasticity is, in this case, much more complicated than the one appearing in Eq. 47 and does not allow the simple factorisation given therein.

---

> > ### Comment · Reviewer_7S1w · 2023-08-19
> > **Re: Rebuttal by Authors**
> >
> > I thank the authors for their detailed response and appreciate the addition of results for the Bayes-optimal estimator. This leads me to increase my score.

---

### Official Review · Reviewer_X846 · 2023-07-07

**Soundness:** 3 good
**Presentation:** 3 good
**Contribution:** 2 fair
**Rating:** 5
**Confidence:** 2

**Summary:**

This paper investigates the asymptotic behavior of Generalized Linear Models (GLM) when the number of training samples $n$, and the dimension of the feature-space $d$ both go to infinity, but the ratio $n/d$ is fixed to some known bounded value $\alpha$. Moreover, authors assume the training data points are drawn from a mixture of two heavy-tailed distributions, which is different from the usual Gaussian assumption in most of the existing works (the mentioned heavy-tailed distributions are constructed by combining uncountably infinitely many Gaussian distributions).

Paper claims to achieve a non-trivial asymptotic characterization of this problem setting, and also try to validate it via a number of experiments on synthetic data. Paper has a number of shortcomings, therefore my current vote is borderline reject. Presentation of the main results needs to be significantly improved, and also I would like to see the comments from other reviewers with more expertise in this particular field to assess the level of technical contribution in this work.

**Strengths:**

- Paper is well-written (at least in most parts), and the literature review part in the introduction section is very informative.
- I have not completely checked the proofs, however, I have not noticed any mathematical mistakes. The technical validity of the theoretical part looks fine. I have not checked the experimental parts.

**Weaknesses:**

- All the theoretical derivations are based on asymptotics, while any result in the non-asymptotic case would be far more interesting.
- I suggest presenting the results more formally, i.e., in the form of Theorems, Lemmas, and etc. Otherwise, the actual level of technical contribution in this work becomes hard to assess. Right now, there are no theorems inside the manuscript. Also, the explanations from L.130 to L.158 are vague (please see the questions section).
- The main motivation behind this work is to assume non-Gaussian distributions (with a possibly infinite covariance) as the components of the mixture model which generates the data. I am concerned with how much this setting would look imporntant and/or interesting to the community. Due to the Gaussian universality principle, the analysis based on the Gaussian assumptions applies (more or less) to all "Gaussian-like" distributions (which covers almost all distributions with bounded moments) as well. Heavy-tailed distributions with power-law tails which do not have a bounded covariance are of course excluded from this list, but how important are they? IMO, authors have not given enough motivation regarding this issue.
- The process which is used to generate the above-mentioned heavy-tailed distributions is very specific: superposition of an uncountably many Gaussians, or equivalently assuming that the covariance matrix of the Gaussian itself is a R.V. with an inverse-Gamma distribution. Authors have not discussed the limitation of this process. How general is it? does it include almost all heavy-tailed distributions?
- I have not completely checked the proofs in supp. However, the mathematical tools used for deriving the results are not sophisticated. Not using sophisticated math or not relying on existing elegant theorems is fine, as long as an important problem has been solved or an interesting discovery has been made. This again takes us to a previous comment, on the importance level of this problem setting. I am not familiar with this particular line of research, so I have to wait for other reviewers to comment on that.

**Questions:**

L.130 to L.158: Results are not properly presented. I suggest using a formal theorem and a set of lemmas.

L.139, Eq (4): What are $\boldsymbol{g}$ and $\boldsymbol{h}$?

L.140, Eq (5): What are $h_{\pm}$, $\omega_{\pm}$, $q$ and ... Actually this list can go on.

The main theoretical contributions are presented in Eq (8) and Eq (9). However, the vague explanation preceding them, would impose a huge negative impact on the potential reader.

**Limitations:**

-

---

> ### Author Rebuttal · Authors · 2023-08-08
>
> We thank the referee for her/his time and comments. Here are some observations concerning the points raised in the report:
> * We agree on the fact that the non-asymptotic behavior is also an interesting problem to consider. In this paper, we have worked in line with a large body of literature that focuses on the asymptotic behavior, however, finite-size *corrections* to the asymptotic behavior are also accessible with the adopted approach (although quite cumbersome) by taking into account corrections to the saddle-point approximation [see, for a review, Lucibello, arXiv:1502.02471]. To our knowledge, this is actually an interesting problem that has not yet been investigated even within the (a priori much simpler) pure Gaussian case. We will keep this suggestion in mind for a future investigation and we thank the referee for raising this point.
> * We have re-organized the material highlighting our main results throughout the text. The set of fixed point equations (7) has been obtained using the heuristic replica method in Appendix A, which we are confident could be made fully rigorous via standard techniques such as Approximate Message Passing (AMP) [Loureiro et al., 2021] or Gordon Minimax techniques [Mignacco et al., 2020]. We will clarify this point in the text with references to the original works. Note that rigorous proof may well require finer technical assumptions. All the following results in our work are instead obtained through fully rigorous derivations from the result in Section 2. We hope that the new form improves the readability of the manuscript.
> * We thank the referee for giving us the opportunity to clarify this important point. The goal of the paper is to “challenge” the validity of a Gaussian universality principle by proposing a model that is under full analytical control and includes the possibility of non-Gaussian datasets. On the one hand, our contribution covers the case of distributions with infinite covariance: such distributions are clearly out of the applicability range of a possible Gaussian universality principle, as correctly stated by the referee. We present this result in this case as a powerful byproduct of our analysis, as, typically, the available theoretical works impose some constraints on the finiteness of the moments, whereas we have no such condition. On the other hand, with respect to the Gaussian universality literature, our most surprising result concerns the case of distributions with *finite* second moment. In the examples of Section 3.1 and Section 3.5, in particular, the existence of moments is controlled via the parameter $a$ of the variance distribution, so that the data distribution has an unbounded k-th moment if  $k\geq 2a$. Crucially, in such cases Gaussian universality breaks down due to the presence of fat tails, i.e., a Gaussian approximation of the dataset (obtained by matching first and second moment) *does not* reproduce the correct asymptotics. We thus *analytically* show that the performances in terms of generalisation do depend on higher moments. We have further clarified this point in our *Introduction*.
> * We would like to thank the referee for raising this important point. It is true in general that any distribution can be approximated (in the L1 sense) by a possible uncountable superposition of Gaussians, whose means and covariances are given according to some law (see, e.g. [Alspach and Sorenson, IEEE Trans. Autom. Control, 17(4), 439-448, 1972]). We leverage this powerful result by considering the case of a law on the second moment of a Gaussian distribution (as a result, the generated functions are even in $\boldsymbol x$); the resulting family is large enough to allow the analysis of *any* tail behavior which was a central goal of the paper. We have added a comment with respect to this in the *Introduction*, with new due references. Moreover, the idea of constructing distributions by taking their parameters as random variables themselves appeared in various disciplines, albeit is known under different names: if in statistical physics it is known as superstatistics [Beck, *Recent developments in superstatistics* 2008], there is also a considerable line of work in Bayesian modeling regarding hierarchical priors and models [Gelman and Hill, *Data Analysis Using Regression and Multilevel/Hierarchical Models*, 2006; Gelman et al., *Bayesian Data Analysis*, 2013], while in probability and statistics such distributions are known as compound probability distributions [Robbins, *Asymptotically Subminimax Solutions of Compound Statistical Decision Problems*, 1985], or as doubly-stochastic models in stochastic processes in particular [Pinsky and Karlin, *An Introduction to Stochastic Modeling*, 2010; Schnoerr et al., *Cox process representation and inference for stochastic reaction-diffusion processes*, 2016]. Such superpositions of distributions are also readily used in direct applications to describe non-Gaussian data in quantitative finance [Delpini and Bormetti, *Minimal model of financial stylized facts* 2011; Langrene et al., *Switching to non-affine stochastic volatility: A closed-form expansion for the Inverse Gamma model*, 2015] or econometrics models [Nelson, *ARCH models as diffusion approximations*, 1990].
> * We have carefully re-read and revised the manuscript, and restructured the presentation of our results. The quantity $\boldsymbol g$ is defined in Eq. 6, right after its appearance in Eq. 4, whilst $\boldsymbol h$ is defined in line 141 via the quantities $h_\pm$ and $\omega_\pm$ introduced by Eq. 5 (in the new version of our manuscript we have now used the notation $\coloneqq$ to clarify that we are indeed defining the quantities therein). Similarly, all order parameters are defined as the solutions of Eq. 7. We hope that the new version of the manuscript will be more clear, and we thank the referee for pointing out places where the clarity of the text could be improved.

---

> > ### Comment · Reviewer_X846 · 2023-08-19
> >
> > I would like to thank the author(s) for their detailed response. After reading the rebuttal and also other reviewers' discussions I decide to slightly raise my rating, but reduce my confidence score.

---

### Official Review · Reviewer_26qY · 2023-07-10

**Soundness:** 3 good
**Presentation:** 1 poor
**Contribution:** 3 good
**Rating:** 6
**Confidence:** 3

**Summary:**

The paper is focused on the non-Gaussian mixture model and asymptotical investigation of the asymptotic characterization of the statistics of the empirical risk minimization estimator. The paper takes under consideration the models with two clusters and applies their analysis to the convex loss functions and regularizers. The empirical evaluations investigate the theoretical aspects in practice.

**Strengths:**

- The theoretical analysis of generalized linear models is an exciting direction, especially in the contents of high-dimensional data.
- The empirical evaluation of synthetic use cases seems to confirm the theoretical investigations.


**Weaknesses:**

- The paper is chaotic and very difficult to follow. It makes the work very difficult to understand. The listed contusions are not defined to the point. They are focused on studying and analyzing, not on practical outcomes. The work is not even summarized (taking into account that there is still some space in the manuscript, it seems to be strange) in the conclusions section, and limitations are not discussed well. Some explanations of crucial symbols are missing in the paper, and the motivations behind some steps are not explained well.

- The proposed theoretical investigation is limited to the models with two classes. How can the results scale to multiclass scenarios?

- The empirical evaluation is limited only to artificial cases. The problem investigated by the authors is very practical, and it is crucial to provide some empirical evaluation using real datasets.

- There are many ways to go beyond the Gaussian distribution. Normalizing flows may be used to model the distributions for each of the considered clusters as an alternative to this approach. It would be beneficial to discuss this issue in the paper and even provide some empirical comparison to the approach.

**Questions:**

Please refer to the remarks from the Weaknesses section.

**Limitations:**

The paper is not organized properly, it is problematic to identify contributions and a strong plot in the work. Empirical evaluation on real cases is missing.

---

> ### Author Rebuttal · Authors · 2023-08-08
>
> We would like to thank the referee for her/his time in reading and evaluating the manuscript. As a general comment, we would like to stress that the goal of our work was to provide, for the first time, a theoretical model to analytically handle the asymptotic properties of classification estimators on non-Gaussian mixtures, in such a way that the effect of non-Gaussianity can be kept *fully* under control and compared with equivalent performances on Gaussian models.
>
> Although the framework of the analysis is therefore, at least at this stage, purely theoretical, it provides a number of important insights: it sheds light on the validity of recent Gaussian universality claims going beyond a large literature which was, up to now, limited to the purely Gaussian case as stated in our *Introduction*.  The insights include, but are not limited to, analytical expressions for the generalisation, training errors, and training loss; analytical formulas for the Bayes-optimal performance and data separability threshold (maximum number of samples possible to perfectly interpolate for a given dimension); role of the regularisation strength; validity of Gaussian universality principle on structured and random-labeled datasets. All these results were determined on a very large class of non-Gaussian distributions, with full control over the fatness of tails and even (non-)existence of data covariance.
> * We would like to thank the referee for the feedback: we revisited the manuscript and improved the readability, taking into account all comments. In particular, we have added a *Conclusions and perspectives* section in which the results have been summarised. We have also discussed more clearly the limitations with respect to the treatment of real datasets in that section. There, we highlight that the main difficulty, in this case, is the choice of the best distribution $\varrho$ given the observed dataset, a problem that however has a long tradition in the context of Bayesian estimation [Alspach and Sorenson, *Nonlinear Bayesian estimation using Gaussian sum approximations*, 1972; Gelman et al., *Bayesian Data Analysis*, 2013]. In a more simplistic approach, it can be observed that, in the case in which the square loss is adopted, the self-consistent equations depend on $\mathbb E[(1+v\Delta)^{-1}]$ and $\mathbb E[(1+v\Delta)^{-2}]$ only, and these quantities can be numerically estimated from the dataset. The exact evaluation of the quality and limitations of such an approximation on a real dataset are left for future investigation. We would like to stress, however, that taking parameters of distributions themselves as random variables, resulting in superpositions of distributions are readily used in direct applications to describe non-Gaussian data in quantitative finance [Delpini and Bormetti, *Minimal model of financial stylized facts*, 2011; Langrene et al., *Switching to non-affine stochastic volatility: A closed-form expansion for the Inverse Gamma model*, 2015] or econometrics models [Nelson, *ARCH models as diffusion approximations*, 1990], so there already exist schemes for convenient choices in for some types of datasets.
> * The theoretical model can be easily generalised to the case of $K$ classes: in the main text, we limited ourselves to the case of 2 classes for the sake of simplicity and clarity. However, to support this answer, we have modified, in Appendix A, the derivation of our results to include the case of $K$ clusters with scalar labels. Other variations (eg, one-hot-encoding labeling) can be obtained following the formalism in the cited reference [Loureiro et al., 2021].
> * The restriction to synthetic datasets is due to the following reason: the provided theoretical prediction relies on the knowledge of the distribution $\varrho$ which, in the case of empirical datasets, can be estimated but is in general not known (see also above). One experiment we would like to perform in the future is indeed the comparison of the results of numerical experiments on real datasets with theoretical predictions obtained after an empirical estimation of $\varrho$, in the spirit of the analysis in [Loureiro et al., 2021] for the case of Gaussian mixtures. On the other hand, as mentioned above, our focus in this contribution was the fact that it is indeed possible to have a simple model within a non-Gaussian setting fully under control and observe therefore a breakdown of Gaussian universality results in it.
> * Although normalising flows are definitely a way to produce non-Gaussian distributions and possibly model a non-Gaussian dataset, as mentioned above the goal of our paper is actually not to provide a tool to model datasets but rather an exactly solvable model, such that the presence of non-Gaussian features can be taken into account and exactly treated. Note that the superposition of the variance distribution with a Gaussian data distribution specifically, while generating a very large family of distributions including ones with any power-law tail, is necessary to employ our analytic method, where Gaussian integration can be conveniently performed by virtue of this data construction starting from Gaussian. We are not aware of references where high-dimensional asymptotics on non-Gaussian distributions is analytically obtained via normalising flows, and we would be extremely grateful to the referee if she/he could point us to some pertinent references, which we will gladly add to the manuscript.

---

> > ### Comment · Reviewer_26qY · 2023-08-18
> > **Thank you for rebuttal**
> >
> > I would like to thank the authors for the clarification during the rebuttal. I read the paper one more time, as well as other reviews and comments. After clarification, I appreciate the theoretical contribution of this work. I still think that the paper requires some rewriting to make it more accessible to the larger community. Moreover, I think that empirical evaluation of real cases is possible at least as a showcase for theoretical considerations. I decided to raise my score.

---

### Official Review · Reviewer_K5kC · 2023-07-19

**Soundness:** 4 excellent
**Presentation:** 3 good
**Contribution:** 3 good
**Rating:** 7
**Confidence:** 4

**Summary:**

The paper derive a theory for training and generalization error when classifying a large number of points from a non-Gaussian high-dimensional data distribution. The data model is a double-stochastic process where a parameter is sampled from a scalar distribution and then a sample is taken from a Gaussian distribution with this parameter as variance. A self-consistent mean-field theory is provided for the case where the number of points is large and proportional to the dimensionality and the equations can be numerically solved for logistic and square losses. The theory is applied to data-sets with finite and infinite covariance, to study the role of regularization, and to estimate the separability threshold for such data. This highlighting both cases of “Gaussian universality” where the results coincide with previous “Gaussian” literature and deviation from such universality.

**Strengths:**

 *	Originality: the tasks and methods are not new, and previous contributions are very well introduced. The originality of the work lies in the successful calculation of the theory for the non-Gaussian case, which is a valuable contribution. Furthermore, the work provides a basis for analyzing when an extrapolation beyond the Gaussian case is justified.
 *	Significance: the paper is important in highlighting where non-Gaussian data may diverge from the Gaussian case discussed in the literature, and as such it opens a venue for future work to use non-Gaussian analysis of real-worlds data, which is an important direction. The conclusions about test error in Gaussian vs non-Gaussian cases is non-trivial (the inversion between figure 1+2 and 3) and as is the finding or optimal finite regularization value for the non-Gaussian case.
 *	Clarity: the paper is in general well-written and can be served as exemplar for providing complicated theoretical results without sacrificing the clarity of the ideas.
 *	Quality: the paper seems technically very sound, with impressive combination of theory and simulations.


**Weaknesses:**

 *	Clarity: some of the notations and ideas presented are only hastily introduced, with two prominent examples being “Superstatistical Features” (from the title) and “uncountable superposition” (from the abstract, which seem overly complicated. To me, a presentation through “double stochastic” process (as in my summary) is straight-forward and require no extra jargon.
Another avenue for improving the understanding of the reader may lie in “Quadratic loss with ridge regularisation” where results are more amendable to interpretation. The authors should have provided more intuition for those results and furthermore point out where does non-Gaussian enters in the self-consistent equations (i.e., what part is shared with the Gaussian case).
 *	Originality: the main part of the work focus on reproducing known results from Gaussian literature and exploring the deviation from them for the non-Gaussian case. In that sense, there is no originality in this work beyond the (impressive) achievement of providing a theory which describe this non-Gaussian case.
 *	Significance: the work would have been more influential if it provided new tools which can be applied to datasets, where a small number of shape parameters is fit to non-Gaussian data and the ability to classify this data can be predicted from theory and then compared to actual classification of the data.
Cases where the Gaussian case predicts the behavior for the non-Gaussian case might deserve a fuller theoretical analysis, perhaps through the analysis suggested for “Quadratic loss with ridge regularisation” above.


**Questions:**

 * Why bother with the classifier estimator phi? Is there any reasonable choice beyond sign?
 * Why do you refer to z* as “the matrix”?
 * Are all the 8 (or 10) order parameters scalars?
 * Can you clarify the interpretation of the proximal h and g? Do their distribution is a mean-field version of some real-worlds quantity?


**Limitations:**

The authors adequately addressed the limitations of their work. Those include the diagonal structure of the covariance matrix (conditional on the value of delta), the use of K=2 which leads to lack of discussion about the mean of the distribution (because they do not affect anything for K=2 beyond their norm), and the resulting theory solvable only numerically.

---

> ### Author Rebuttal · Authors · 2023-08-08
>
> We would like to thank the reviewer for her/his remarks and positive evaluation of our paper, and for capturing the spirit of our contribution very well. We are grateful to the referee for her/his suggestions about improving the clarity of the manuscript, which we implemented in the new version. We also thank her/him for suggestions regarding possible applications of the theory, in particular, in its simplest form given in the case of ridge regression: we have added a comment, in this sense, in a newly introduced *Conclusions and perspectives* section.
>
> We list our answers to her/his questions below, hoping that they will be satisfactory.
> * The referee is right about the fact that, in the given setting, the sign function is indeed the only reasonable choice, and the one we indeed adopted through the text. We decided to leave the classifier more general as, in the new version, we have extended the replica calculation to the multiclass case in the Appendix, where the classifier’s choice is less obvious.
> * Thanks for pointing out the typo regarding $\boldsymbol z^\star$, which is, indeed, a vector!
> * All order parameters are scalars: in the paper, we tried to consistently use bold fonts for vectorial/matricial objects (e.g., $\boldsymbol g$) and normal fonts for scalars.
> * The proximals $h_\pm$ can be seen as expressing the statistics of the preactivation $\frac{\boldsymbol w^\star{}^\intercal \boldsymbol x^\nu}{\sqrt d}+b^\star$ when $\boldsymbol x^\nu$ is a training set datapoint with label $y^\nu=\pm1$. The proximal $\boldsymbol g$, instead, captures the statistics of $\boldsymbol w^\star$ itself (as expressed by Eq. 8).

---

### Author Rebuttal · Authors · 2023-08-08

*General remarks* We thank the reviewers for their helpful feedback which helped us improve the readability and clarity of our work, and better express the importance of our contribution. To take into account their comments, we have prepared a new version of our manuscript, in which, beyond addressing the referees’ comments, we provide additional, important results, namely
* a generalisation of our result to $K$ classes (to appear in the Appendix);
* a derivation of an analytical formula for the data separability threshold;
* a derivation of an analytical formula for the Bayes-optimal error, and the comparison of the bound with the ERM results (we provide, as an example, the new Fig. 1 where the Bayes-optimal error bound is given as a dashed line for each value $a$).

We have answered the questions and addressed specific concerns of each reviewer in the individual answers below.

---

> ### Comment · Reviewer_K5kC · 2023-08-15
> **Response to Author Rebuttal**
>
> I stand by my assessment that this is a good candidate for acceptance. Looking forward to seeing if the other, less positive reviewers reconsider following the improved presentation and additional results.

---

### Decision · Program_Chairs · 2023-09-21

**Decision:**

Accept (poster)

**Comment:**

This paper provides statistical rates for classification in high dimensions, with the main contribution being that the data distribution is a superposition of Gaussians, and thus flexible enough to capture many phenomenon heretofore avoided in this literature, such as heavy tails. I am happy to recommend acceptance, though I urge the authors to adjust the manuscript during the camera ready phase to reflect the fruitful reviewer discussions.